# NMDA receptors control development of somatosensory callosal axonal projections

Jing Zhou[1,2], Yong Lin[1,3], Trung Huynh[1,2], Hirofumi Noguchi[1,2], Jeffrey O Bush[4,5], Samuel J Pleasure[1,2,6]*

[1]Department of Neurology, University of California, San Francisco, San Francisco, United States; [2]Weill Institute for Neurosciences, University of California, San Francisco, San Francisco, United States; [3]Department of Neurological Surgery, Ren Ji Hospital, School of Medicine, Shanghai Jiao Tong University, Shanghai, China; [4]Department of Cell and Tissue Biology, Program in Craniofacial Biology and Institute for Human Genetics, University of California, San Francisco, San Francisco, United States; [5]Eli and Edythe Broad Center of Regeneration Medicine and Stem Cell Research, University of California, San Francisco, San Francisco, United States; [6]Programs in Neuroscience and Developmental Stem Cell Biology, Eli and Edythe Broad Center of Regeneration Medicine and Stem Cell Research, Kavli Institute for Fundamental Neuroscience, San Francisco, United States

**Abstract** Callosal projections from primary somatosensory cortex (S1) are key for processing somatosensory inputs and integrating sensory-motor information. How the callosal innervation pattern in S1 is formed during early postnatal development is not clear. We found that the normal termination pattern of these callosal projections is disrupted in cortex specific NMDAR mutants. Rather than projecting selectively to the primary/secondary somatosensory cortex (S1/S2) border, axons were uniformly distributed throughout S1. In addition, the density of this projection increased over postnatal life until the mice died by P30. By combining genetic and antibody-mediated loss of function, we demonstrated that it is GluN2B-containing NMDA receptors in target S1 that mediate this guidance phenotype, thus playing a central role in interhemispheric connectivity. Furthermore, we found that this function of NMDA receptors in callosal circuit formation is independent of ion channel function and works with the EPHRIN-B/EPHB system. Thus, NMDAR in target S1 cortex regulates the formation callosal circuits perhaps by modulating EPH-dependent repulsion.

*For correspondence:
samuel.pleasure@ucsf.edu

Competing interests: The authors declare that no competing interests exist.

## Introduction

Synaptic connections between neurons form circuits that can convey neural information. Abnormalities at any stage of synaptic circuit development can result in neuropsychiatric pathology. The corpus callosum (CC) is the largest interhemispheric commissural circuit in mammals. The connectivity of the CC is essential for coordinated sensory-motor function and for many higher cognitive processes, and CC pathology is implicated in a variety of developmental disorders (*Paul, 2011*).

Callosal projections originate from pyramidal neurons located in layers II/III, V, and VI and traverse the CC to form synapses with neurons in contralateral homotopic or heterotopic cortical areas. We previously showed (*Zhou et al., 2013*) that the medial-lateral topography of callosal neurons in the cortex is tightly constrained by the dorsal-ventral (D-V) position of axons within the CC. The axon position within the CC determines its terminal location in the contralateral cortex, with dorsally located axons projecting medially and ventrally located axons projecting laterally. As such, the spatial organization of topographically represented information from one hemisphere is

preserved as it is projected onto the contralateral hemisphere. However, the molecular determinants regulating proper targeting of commissural projections remain unknown.

In vivo $Ca^{2+}$ imaging and multiunit recordings show distinct patterns of neural activity in the cortex of newborn mice (*Adelsberger et al., 2005*; *Khazipov and Luhmann, 2006*; *Khazipov et al., 2004*). These activity patterns synchronize spatially and temporally distinct neural networks and may play important roles in wiring cortical maps (*Allène et al., 2008*; *Golshani et al., 2009*; *Yang et al., 2009*). Suppressing endogenous neural activity by overexpressing the inward rectifying potassium channel Kir2.1 in callosal neurons delays axon growth and results ultimately in layer-targeting defects in visual cortex and somatosensory cortex (*Mizuno et al., 2007*; *Rodríguez-Tornos et al., 2016*; *Wang et al., 2007*). Sensory deprivation by ablating whiskers or transecting the infraorbital nerve (ION) before P5 blocks sensory activity to the trigeminal nerve and disrupts barrel field formation in primary somatosensory cortex (S1) with secondary disruption of callosal innervation at the S1/S2 border (*Huang et al., 2013*; *Suárez et al., 2014*). These studies show that directly reducing neural activity or blocking ascending sensory inputs to callosal neurons affects callosal targeting and map formation. However, the molecular mechanisms governing these events are not clear.

Neural activity is generally propagated from the axons of presynaptic neurons to the dendrites of postsynaptic neurons by the secretion of neurotransmitters. Neurotransmitter receptors located on the postsynaptic neuron regulate synaptic transmission. The NMDA receptor (NMDAR) is a glutamatergic neurotransmitter receptor located at the synapses that mediates the vast majority of excitatory neurotransmission in the cortex (*Traynelis et al., 2010*). NMDAR mediated synaptic transmission is important in generating synchronized activity patterns in immature cortex, suggesting that NMDAR $Ca^{2+}$ channel may be involved in neural circuit formation.

In this study we examined the role of NMDAR in the formation of callosal circuitry. Initially, our hypothesis was that NMDARs would be important modulators of callosal circuit formation and that this would be mediated through the ion channel function by regulating neural activity. Indeed, we did find a crucial role for NMDAR in regulating callosal innervation patterns. To our surprise, this was not mediated by the ion channel function of NMDAR. Rather, there was a specific role for the GluN2B-containing NMDAR and loss of NMDAR resulted in changes in EPHB2 expression. EPHB2 is known to be necessary for localization of NMDAR to synapses (*Nolt et al., 2011*); however, we found that this requirement was reciprocal – when NMDARs are lost, EPHB2 protein expression during development is lost as well. Most importantly, this is the first demonstration that interactions between NMDAR and EPHRIN-B/EPHB are required for neural circuit formation during development.

## Results

### Postnatal development of the S1 callosal projection

The development of commissural S1 projections serves as an ideal model to study interhemispheric circuit development. To understand normal development of the S1 callosal projection, we labeled the progenitor cells for layer II/III neurons with enhanced green fluorescent protein (EGFP) by in utero electroporation at embryonic day (E) 15.5 and examined callosal development at four critical time points (*Figure 1*). At postnatal day (P) 5, the callosal axons from S1 had reached the white matter underneath contralateral S1 (*Figure 1B*). At P8, the callosal axons were diffusely distributed in contralateral S1 (*Figure 1C*). By P12, pruning of excess projections led to a refined innervation pattern with a narrow band limited to the S1/S2 border (*Figure 1D*). After P12, the pattern is generally stable, as shown at P30 (*Figure 1E*).

### GluN1 knock-out (KO) mice have disrupted callosal innervation

NMDARs are heteromeric receptor channel complexes that differ in subunit composition. To date, seven different subunits have been identified: the GluN1 subunit, four distinct NR2 subunits (A–D), and a pair of GluN3 subunits. The GluN1 subunit is the essential subunit of NMDARs (*Dingledine et al., 1999*; *Paoletti et al., 2013*). Global GluN1 knockout mice die within a few hours of birth (*Forrest et al., 1994*). To explore the role of NMDAR in callosal development, we generated cortex-specific GluN1 knock-out (KO) mice by crossing a floxed GluN1 allele mice (*Grin1^{fl/fl}*) with Emx1-Cre recombinase (Cre) mice (*Emx1^{cre/+}*) thereby selectively deleting GluN1 in excitatory

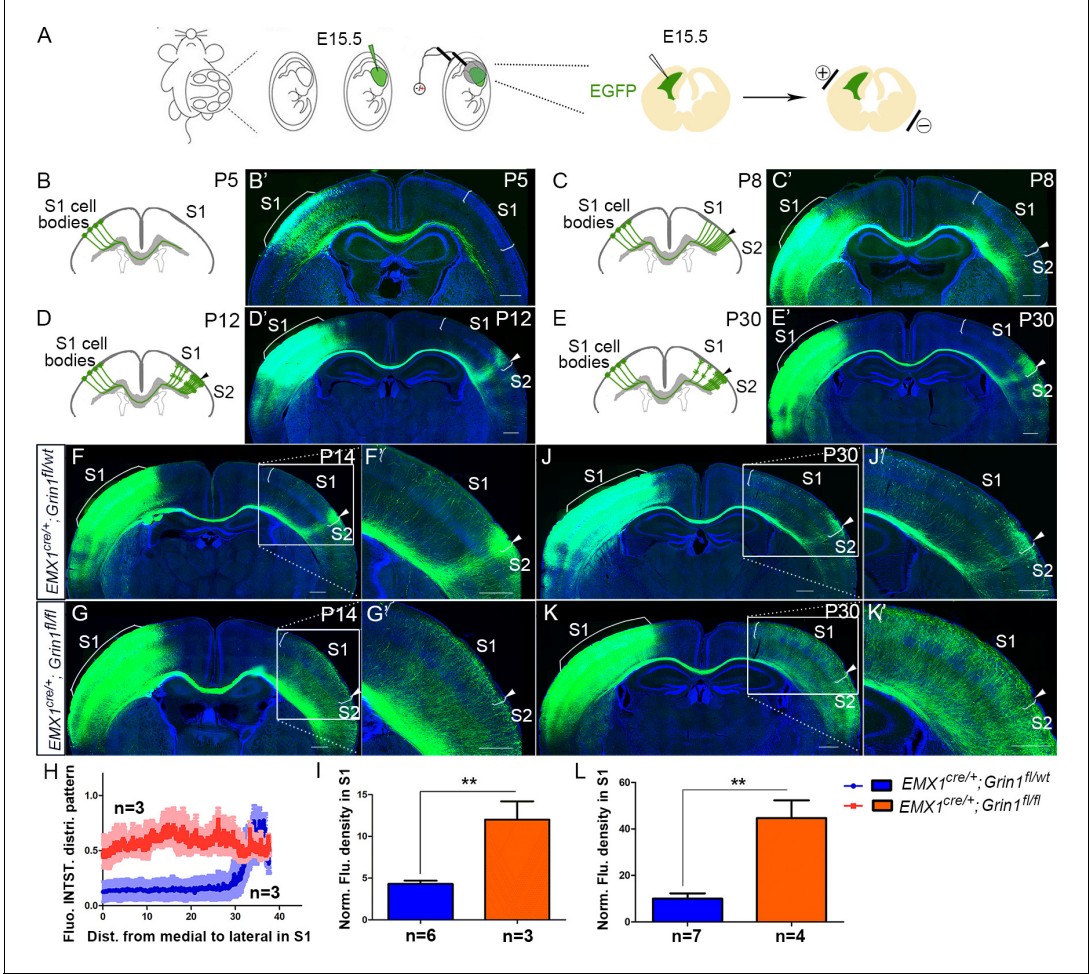

**Figure 1.** The callosal somatosensory innervation pattern was disrupted in *Emx1^cre/+; Grin1^fl/fl* mice. (A–E) Postnatal development of callosal projection in S1. (A) EGFP plasmid injected into lateral ventricle of embryo at embryonic day 15.5 (E15.5) and electrical pulse given to enable the plasmid to enter cortical progenitor cells of layer II/III in the ventricular zone. (B, B') At postnatal day 5 (P5), the callosal axons from S1 had reached the white matter underneath contralateral S1. (C, C') At P8, the callosal axons were diffusely distributed in contralateral S1. (D, D') By P12, pruning of excess projections led to a refined innervation pattern with a narrow band limited to the S1/S2 border. (E, E') After P12, the pattern was stable as observed at P30. (F) In P14 control mice (*Emx1^cre/+; Grin1^fl/wt*), the callosal innervation pattern of S1 of the contralateral cortex is well differentiated with a dense innervation at S1/S2 border. The pattern persists to P30 (J). (G) In GluN1 KO mice (*Emx1^cre/+; Grin1^fl/fl*), the innervation pattern was disrupted and projections were extremely diffused which also persisted to P30 (K). (H) Quantification of fluorescent intensity across the medial to lateral extent of the S1. (I, L) Quantification of fluorescence density of S1 region of control vs. GluN1 KO mice at P14 (I, p=0.002) and P30 (L, p=0.0003). Scale bar: 500 µm for all images. S1: primary somatosensory cortex; S2: secondary somatosensory cortex.

The online version of this article includes the following figure supplement(s) for figure 1:

**Figure supplement 1.** The expression of NMDAR in cortex was reduced in *Emx1^cre/+; Grin1^fl/fl* mice.

cortical projection neurons. Since GluN1 is essential, GluN1 deletion results in loss of functional NMDARs in cortical excitatory neurons. Immunostaining and western blot at P8 showed that expression of cortical GluN1 was greatly diminished (*Figure 1—figure supplement 1*) in GluN1 KO mice. Consistent with previous studies (*Iwasato et al., 2000*; *Lo et al., 2013*), the organization of thalamo-cortical barrels in Layer IV of S1 was somewhat disrupted but still apparent in GluN1 KO mice at P8, as revealed by vesicular glutamate transporter 2 (VGlut2) immunostaining (*Figure 1—figure supplement 1*).

To investigate whether NMDAR plays a role in the targeting of callosal projections to contralateral cortex, we first examined the callosal innervation pattern at P14, by which point the mature pattern has formed. In littermate controls (*Emx1^cre/+; Grin1^fl/wt*), a dense area of innervation was formed at the S1/S2 border (*Figure 1F*) and the overall pattern was the same as wild-type controls in

*Figure 1D*. However, in GluN1 KO mice (*Emx1^{cre/+}; Grin1^{fl/fl}* mice), the normal restricted pattern of callosal targeting to the S1/S2 border was absent, and callosal axons were uniformly distributed throughout the contralateral somatosensory cortex (*Figure 1G*). Quantitative fluorescence intensity analysis showed that the axon distribution pattern in S1 was statistically significantly different between littermate control and GluN1 KO mice (*Figure 1H*). Additionally, fluorescence density analysis showed that GluN1 KO mice also had more overall callosal axons innervating S1 at P14 (*Figure 1I*). This result suggests that the NMDAR plays a role in callosal circuit formation. In the absence of NMDAR, the targeted callosal innervation of the S1/S2 border was lost, and the overall callosal innervation in S1 was significantly increased. GluN1 KO mice were smaller than littermate controls after P5 (data not shown) and rarely survived past P30 so we chose P30 as the last time point to determine if this defect persists. We found that this phenotype worsens after P14 (*Figure 1J–L*).

## GluN1 KO mice prematurely innervate S1

To determine when this prominent targeting defect can first be detected, we examined different time points (P0, P3, P5, and P6) corresponding to critical phases of initial CC circuit formation – initial axon extension to ipsilateral CC (P0), axons crossing the midline (P3), axons reaching the white matter underneath contralateral S1 (P5), and axons starting to innervate S1 (P6). GluN1 KO mice showed no differences with littermate controls at P0, P3, and P5 (*Figure 2—figure supplement 1*) indicating no difference in the overall rate of axon growth. However, GluN1 KO mice showed earlier and increased innervation of S1 at P6, when callosal axons start entering the contralateral cortex

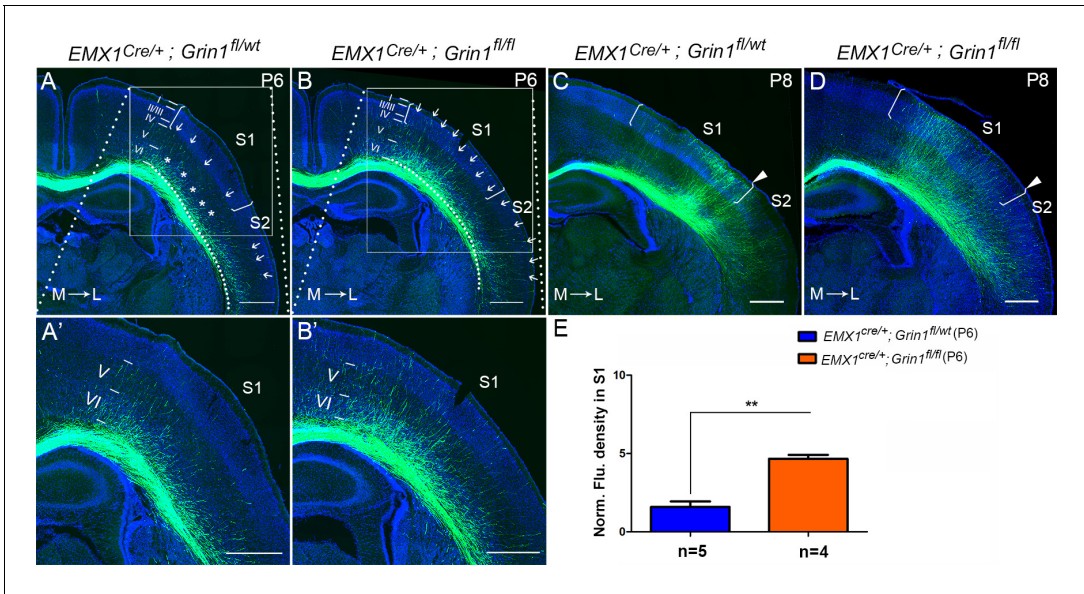

**Figure 2.** The callosal innervation defect was first detected at P6 in *Emx1^{cre/+}; Grin1^{fl/fl}* mice. (A, A') At P6, most axons in control grew into deeper layer VI of S1 (see '*'); a few axons grew to layer V from medial to lateral S1 (see arrows). However, axons projecting to lateral S2 had grown to layer IV which was apparently faster than the axons in S1 (see arrows). (B, B') In GluN1 KO mice, most axons had grown to layer V and some even grew to layer I (see arrows) at P6. (C, D) At P8, axons in control and mutant mice had grown to the superficial layer of cortex. However, the innervation patterns were different. Controls showed more axon innervation in the lateral S1 with dense callosal innervation at S1/S2 border (C). Mutants showed slightly more axon innervation in the medial S1 (D). (E) The fluorescence density of mutant mice in S1 was significantly higher than in control mice at P6 which suggested that the mutants had increased axon innervation in contralateral S1 at P6. p=0.003. Scale bar: 500 µm for all images. The square brackets in all images outline the S1. The arrow heads in all images outline the S1/S2 border. White lines outline different layers in the cortex of **A–D**. M: medial; L: lateral.

The online version of this article includes the following figure supplement(s) for figure 2:

**Figure supplement 1.** There was no difference between *Emx1^{cre/+}; Grin1^{fl/wt}* and *Emx1^{cre/+}; Grin1^{fl/fl}* mice during axonal extension into the ipsilateral CC (P0) and to the contralateral CC (P5).

**Figure supplement 2.** No increased cell death in S1 of *Emx1^{cre/+}; Grin1^{fl/fl}* mice at P6.

**Figure supplement 3.** Callosal axon density analysis by Image J.

(*Figure 2B and E*). In control littermates, at P6, callosal axons were gathered into a bundle under contralateral S1 with few axons penetrating into S1. The penetrating axons in S1 were distributed across layer VI ('*' in S1 in *Figure 2A*), and layer V (arrows in S1 in *Figure 2A*) but failed to penetrate to more superficial layers. In contrast, axons targeting S1 in GluN1 KO mice at P6 had extended past layer V, with many axons terminating in layer II/III (arrows in S1 in *Figure 2B*). Interestingly, while control mice showed equal degrees of callosal innervation from medial to lateral S1 (see arrows in S1 in *Figure 2A*), GluN1 KO mice showed preferential early callosal innervation of medial S1 (see arrows in S1 in *Figure 2B*), and this was even more apparent at P8 (*Figure 2C and D*).

Because this defect was first detected at P6, we wondered whether increased cell death of target cortical neurons could account for the mis-innervation of S1. We used active (cleaved) caspase-3 as a marker of cell death in control and GluN1 KO mice at P6. In control mice, cleaved caspase-3⁺ cells were mostly detected in layer II/III of primary motor cortex (M1), and rarely were observed in other cortical regions (*Figure 2—figure supplement 2A*). Mutants did have increased cell death in layer II/III of M1 (*Figure 2—figure supplement 2B*'); however, there was no increase in cell death in other cortical regions including S1 (*Figure 2—figure supplement 2B*'').

Taken together, these data indicate that the deletion of NMDAR in excitatory cortical neurons leads to premature and disrupted callosal innervation of contralateral S1. Furthermore, this excessive innervation persists and is not corrected by pruning of mistargeted axons at later developmental stages.

## The NMDAR is required in target neurons for normal callosal innervation

In GluN1 KO mice, the NMDAR is deleted from both projecting (presynaptic) and target (postsynaptic) neurons. To examine the role of NMDAR in presynaptic projecting callosal neurons, we electroporated vectors encoding Cre and EGFP into S1 of homozygous floxed GluN1 mice at E15.5 (*Figure 3A*). Compared with littermate controls (*Figure 3B*), presynaptic deletion of GluN1 (ipsiS1-/-) from neurons had no discernible effect on callosal innervation at P14 (*Figure 3C*). This suggests that NMDAR is not required for projecting neurons to properly target S1. To determine whether GluN1 is required in the S1 target neurons, we deleted NMDAR in contralateral target neurons by in utero electroporation. To delete NMDAR in all cortical layers of the contralateral cortex, a Cre vector was electroporated into *Grin1^{fl/fl}; Rosa26^{fs-tdTomato}* mice in the contralateral/target S1 at E12.5 followed by a second electroporation of EGFP on E15.5 to label ipsilateral/projecting S1 (*Figure 3E*). *Rosa26^{fs-tdTomato}* mice express the fluorescent protein tdTomato after Cre-mediated recombination, and therefore labeled cells where Cre-mediated excision occurred and thus GluN1 should be deleted (*Madisen et al., 2010*). Control mice (*Grin1^{wt/wt}; Rosa26^{fs-tdTomato}* mice) after double electroporation were analyzed at P14 (*Figure 3F*) and showed no defects. However, selective deletion of postsynaptic GluN1 in experimental mice strikingly increased callosal S1 innervation (*Figure 3G and H*, *Figure 3—figure supplement 1*) similar to the deletion of GluN1 in all excitatory cortical neurons (*Figure 1G*). These results suggest that NMDAR is required in the target neurons only for normal callosal circuit formation.

## Increased callosal innervation in S1 after contralateral injection of anti-NMDAR antibodies

NMDAR antibodies directed to an extracellular domain of GluN1 are known to downregulate the numbers of surface NMDARs. To further examine that it is the contralateral/target expression of NMDAR in deep cortical layers is required for callosal axon targeting, we decided to use infusions of anti-GluN1 antibodies to block NMDAR function and expression in a temporally specific manner. We chose a commercial antibody against the amino acid residues 385–399 in the extracellular N-terminal domain of GluN1. It has been shown that this antibody can alter the surface distribution and dynamics of NMDAR (*Dupuis et al., 2014*). To examine the efficiency of injection and whether the injection itself can cause brain damage, we injected the anti-GluN1 antibodies into the lateral ventricle from P2 to P8 and perfused the mice 3 hr after the last injection. The distribution of anti-GluN1 antibodies was most abundant in deep cortical layers of the ipsilateral injected hemisphere (*Figure 4—figure supplement 1*). Thus, presumably due to the widespread high level of expression of GluN1, the

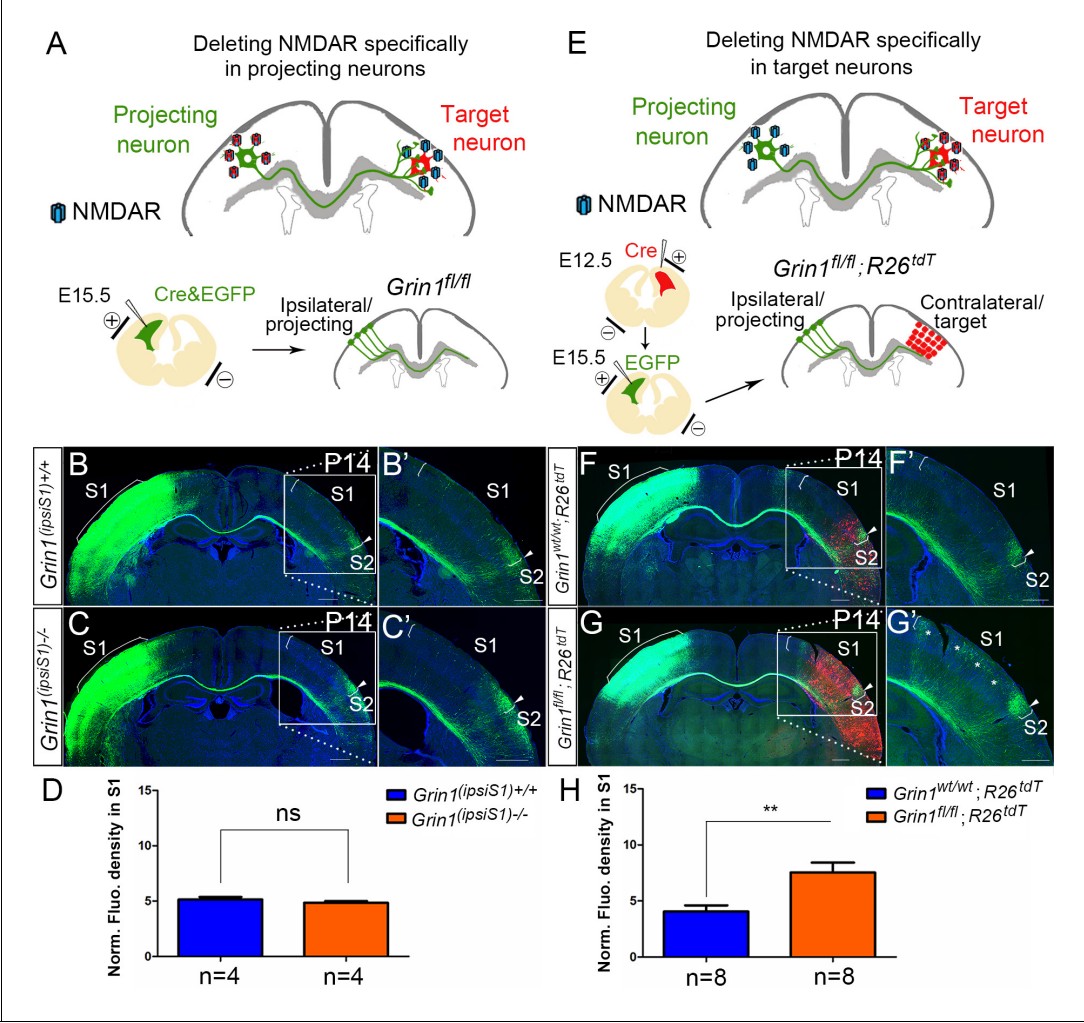

**Figure 3.** NMDAR is required in target neurons for normal callosal innervation. (A–D) Deleting NMDAR specifically in projecting neurons. Vectors expressing Cre-recombinase (Cre) and EGFP were delivered into S1 of floxed GluN1 mice (*Grin1*$^{fl/wt}$ x *Grin1*$^{fl/wt}$) by in utero electroporation at E15.5 (A). Callosal innervation patterns at P14 in control *Grin1*$^{+/+}$ mice (B) and mice after ipsilateral deletion (C). (D) Quantification of fluorescence density. p=0.317. (E–H) Deleting NMDAR specifically in target neurons. GluN1 was deleted in target contralateral S1 by in utero electroporation of Cre at E12.5 in *Grin1*$^{fl/fl}$; *Rosa26*$^{fs-tdTomato}$ mice, the ipsilateral projecting neurons were labeled by EGFP at E15.5 (E). Compared with control *Grin1*$^{wt/wt}$; *Rosa26*$^{fs-tdTomato}$ (F), *Grin1*$^{fl/fl}$; *Rosa26*$^{fs-tdTomato}$ mice which specifically deleted GluN1 in target S1 showed increased callosal innervation in S1 as '*' shows (G). (H) Quantification of fluorescence density. p=0.002. Scale bar: 500 μm for all images. *R26*$^{tdT}$: *Rosa26*$^{fs-tdTomato}$.

The online version of this article includes the following figure supplement(s) for figure 3:

**Figure supplement 1.** Deleting NMDAR specifically in target neurons.

antibodies are not distributed widely in the cortex, but rather remain predominantly on the side of injection.

Next, we injected the anti-GluN1 antibodies into the lateral ventricle from P2 to P12 either ipsilateral or contralateral to the origin of EGFP labeled callosal neurons and examined callosal innervation patterns at P14. The fluorescence density analysis at P14 showed that the contralateral (*Figure 4E–H*) but not ipsilateral antibody injections (*Figure 4A–D*) led to more overall callosal innervation in S1 at P14. This further supports our genetic data that contralateral/target expression of NMDAR in deep cortical layers is required for callosal axon targeting.

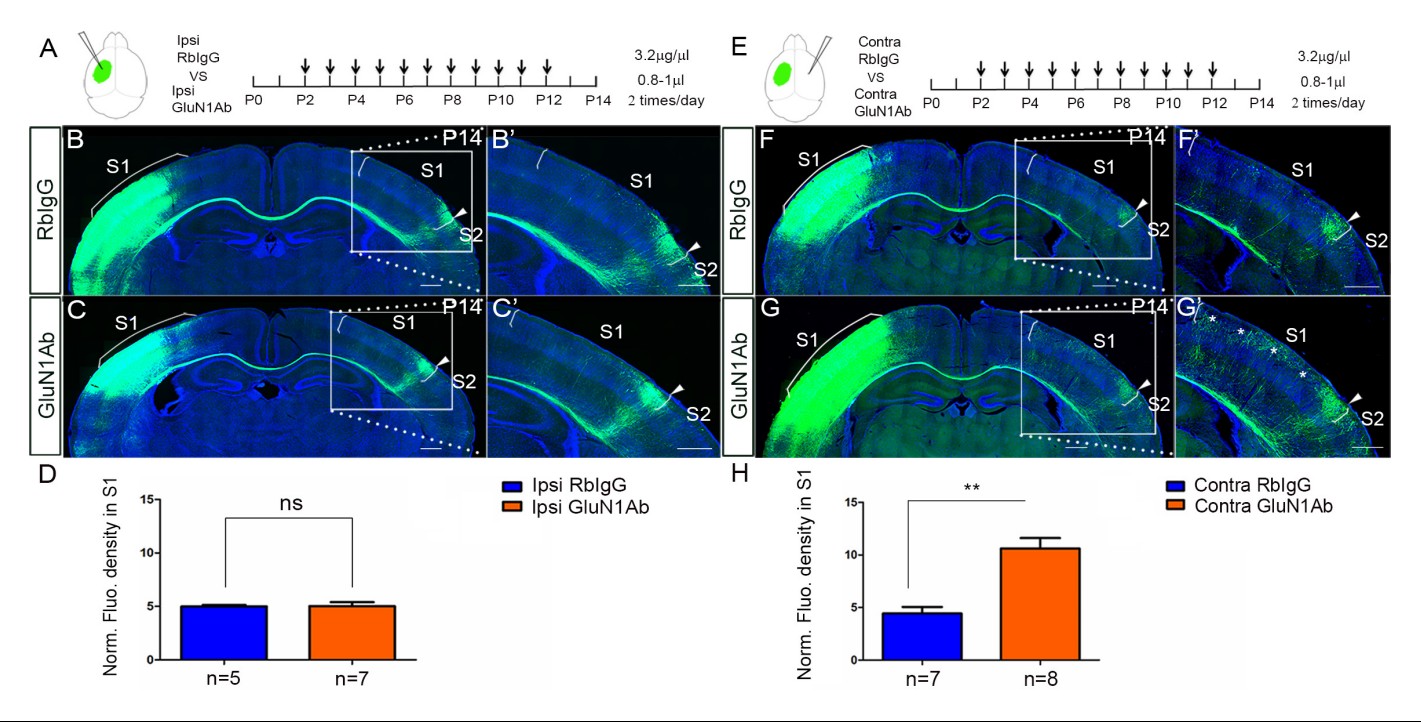

**Figure 4.** Increased callosal innervation in S1 after contralateral but not ipsilateral injection of anti-NMDAR antibodies from P2 to P12. (A–D) Anti-GluN1 antibodies were injected into the lateral ventricle from P2 to P12 in ipsilateral cortex. RbIgG served as control. Compared with control (B), antibody injection in mice did not show increased callosal innervation in S1 at P14 (C). (D) Quantification of fluorescence density. p=0.94. (E–H) Anti-GluN1 antibodies were injected into the lateral ventricle from P2 to P12 in contralateral cortex. Compared with control (F), antibody injection in mice showed increased callosal innervation in S1 at P14 (see '*', G). (H) Quantification of fluorescence density. p=0.0002. Scale bar: 500 μm for all images.

The online version of this article includes the following figure supplement(s) for figure 4:

**Figure supplement 1.** The efficiency of intraventricular antibody injection and the distribution territory in the cortex after 3 hr of last injection.

## NMDAR are required specifically during callosal axon growth into contralateral S1

The antibody injections from P2 to P12 cover all the critical postnatal phases of CC development – (a) axons crossing midline (P3); (b) axons reaching the CC underneath contralateral S1 (P5); (c) growth into S1 (P6–P8); and (d) the refinement of projections (P8–P12). Thus, it is difficult to be certain when during this time the antibodies are acting to cause the observed increased callosal innervation. Our genetic data indicates the role of NMDAR in callosal development is first seen during the process of callosal axon growth into S1. However, the callosal innervation pattern in S1 does not only include the process of projecting into S1 but also the later refinement of projection. To address the temporal role of NMDAR in callosal development, we injected antibodies either from P4 to P8 or from P8 to P14 and examined the callosal innervation pattern at P14. We found that antibody injections from P4 to P8 had increased callosal axonal growth into S1 similar to that observed in NMDAR genetic deletion mice (*Figure 5A–D*). However, antibody injections only from P8 to P14 had no effect on the overall callosal innervation pattern and we saw no increased callosal innervation in S1 (*Figure 5E–H*). As mentioned before, the GluN1 KO mice showed earlier and increased innervation of S1 when the callosal axons started entering target cortex at P6 (*Figure 2B*), but there was no difference between GluN1 KO and littermate controls before P6 (*Figure 2—figure supplement 1*). Thus, taking the genetic and antibody injection data together suggests that the crucial effect of NMDAR on callosal circuit formation is primarily during callosal projection into the cortex (P6–P8).

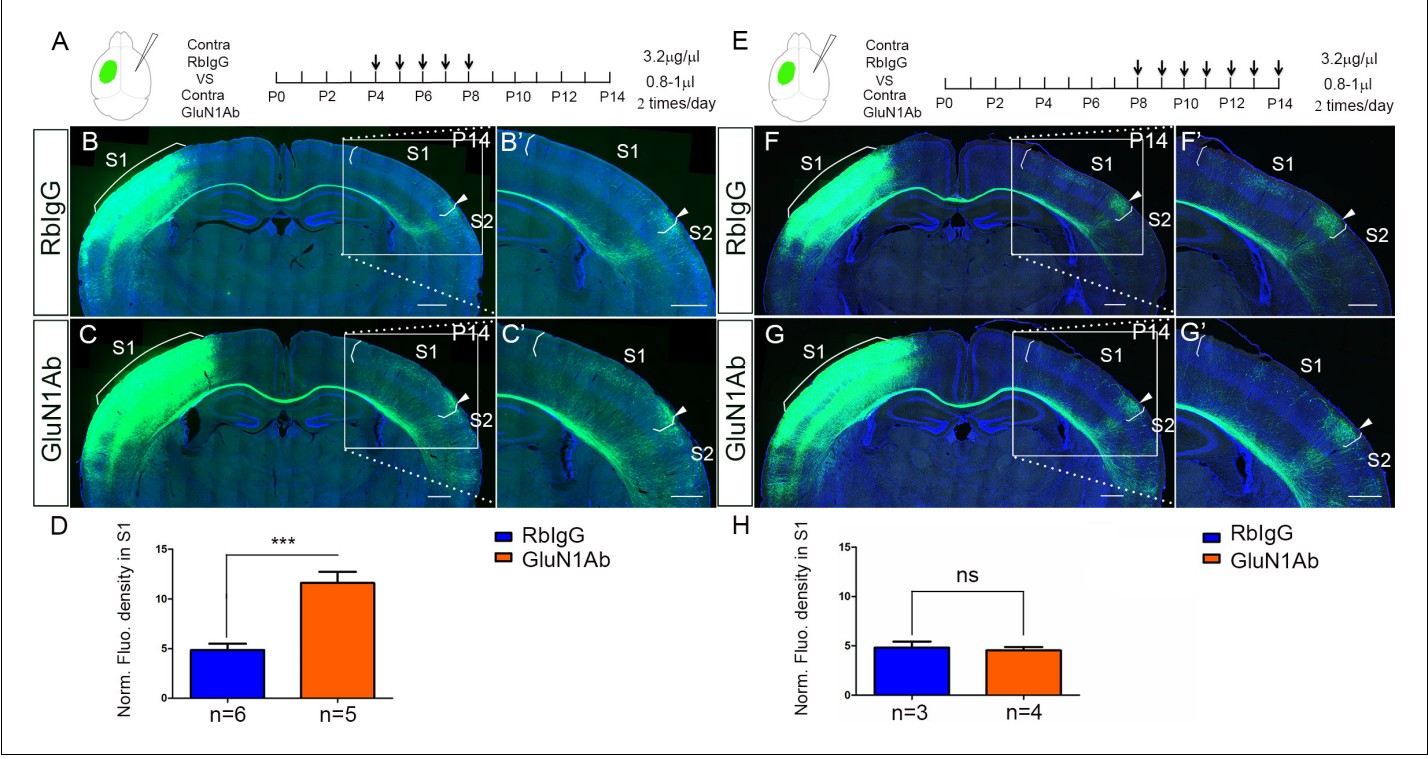

**Figure 5.** Contralateral injection of anti-NMDAR antibodies from P4 to P8 but not P8 to P14 had increased callosal innervation in S1. (A–D) Anti-GluN1 antibodies were injected into the lateral ventricle from P4 to P8 in contralateral cortex. RbIgG served as control. Compared with control (B), antibody injection in mice show increased callosal innervation in S1 at P14 (C). (D) Quantification of fluorescence density. p=0.004. (E–H) Anti-GluN1 antibodies were injected into the lateral ventricle from P8 to P14 in contralateral cortex. Compared with control (F), antibody injection in mice did not show increased callosal innervation in S1 at P14 (G). (H) Quantification of fluorescence density. p=0.69. Scale bar: 500 µm for all images.

## GluN2B, but not GluN2A, is required for callosal axon targeting

GluN1 is an obligatory component of tetrameric NMDA receptors and is required for assembly of functional NMDAR. Therefore, NMDARs were entirely absent from excitatory cortical projection neurons in our GluN1 KO mice. In the forebrain, GluN1 primarily assembles with GluN2A and GluN2B to form functional NMDARs. GluN2A- and GluN2B-containing NMDARs are functionally distinct (*Kutsuwada et al., 1992*; *Loftis and Janowsky, 2003*) and follow different developmental expression trajectories with GluN2B as the major NR2 subunit during the first postnatal week and GluN2A expression present but increasing thereafter (*Liu et al., 2004*; *Monyer et al., 1994*; *Sans et al., 2000*; *Sheng et al., 1994*). We wondered whether GluN2A-containing or GluN2B-containing NMDARs play different roles in S1 callosal development. Thus, we crossed *Emx1*[cre/+] mice with *Grin2a*[fl/fl] mice (GluN2A KO) and *Grin2b*[fl/fl] mice (GluN2B KO) (*Gray et al., 2011*). To address previous studies that found that GluN2A was expressed at lower levels during development, we checked the expression of GluN2A in S1 at P8 and found that it was dramatically diminished in *Emx1*[cre/+]; *Grin2a*[fl/fl] mice (*Figure 6—figure supplement 2A and B*), while the expression of GluN2B was the same in control and *Emx1*[cre/+]; *Grin2a*[fl/fl] mice (*Figure 6—figure supplement 2A and C*), and similarly GluN2A expression was unchanged in *Emx1*[cre/+]; *Grin2b*[fl/fl] mice (**data not shown**).

In GluN2A KO mice, the overall callosal innervation pattern was similar to control mice, although there was increased callosal innervation at the M1/S1 border at P14 (*Figure 6A–C*). The body size of GluN2A KO mice was not significantly different from littermate controls, and they survived to adulthood. At P30, the general callosal innervation pattern of GluN2A KO mice was similar to littermate controls; however, the increased callosal innervation at the M1/S1 border persisted (*Figure 6—figure supplement 1A–C*). In contrast, GluN2B deletion phenocopied GluN1 deletion with increased S1 innervation and loss of targeted innervation of the S1/S2 border at P14 (*Figure 6D–F*). Similar to

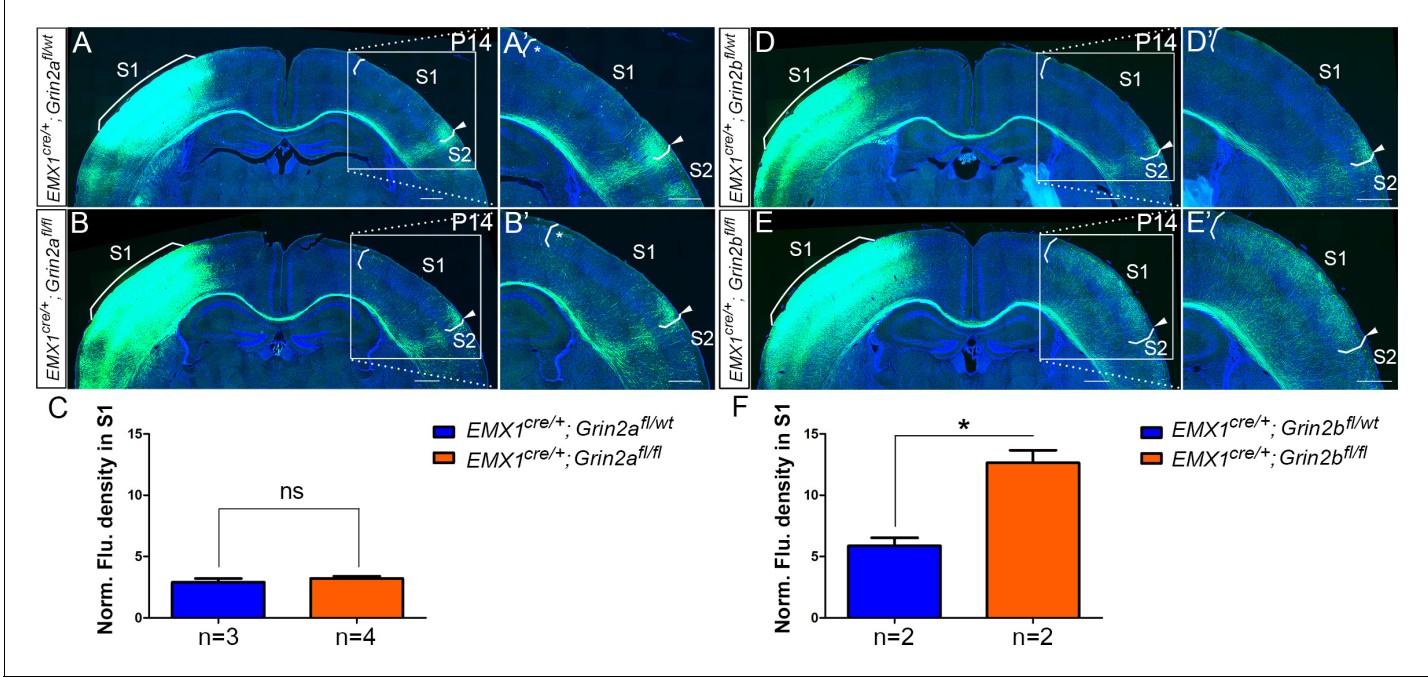

**Figure 6.** $Emx1^{cre/+}$; $Grin2b^{fl/fl}$ but not $Emx1^{cre/+}$; $Grin2a^{fl/fl}$ mice had the same disrupted callosal innervation patterns as $Emx1^{cre/+}$; $Grin1^{fl/fl}$ at P14. Callosal innervation patterns in control $Emx1^{cre/+}$; $Grin2a^{fl/wt}$ (A) and $Emx1^{cre/+}$; $Grin2a^{fl/fl}$ mice (B) at P14. '*' points out M1/S1 border. (C) Quantification of fluorescence density. p=0.392. Callosal innervation patterns in control $Emx1^{cre/+}$; $Grin2b^{fl/wt}$ (D) and $Emx1^{cre/+}$; $Grin2b^{fl/fl}$ mice (E) at P14. (F) Quantification of fluorescence density. p=0.03. Scale bar: 500 μm for all images.

The online version of this article includes the following figure supplement(s) for figure 6:

**Figure supplement 1.** NR2B ($Emx1^{cre/+}$; $Grin2b^{fl/fl}$) but not NR2A ($Emx1^{cre/+}$; $Grin2a^{fl/fl}$) mice had same disrupted callosal innervation patterns as $Emx1^{cre/+}$; $Grin1^{fl/fl}$ at P30.

**Figure supplement 2.** The protein expression level of GluN2A and GluN2B in S1 at P8.

GluN1 KO mice, GluN2B KO mice were smaller than littermate controls after P5 and rarely survived past P30. Like GluN1 KO mice, this phenotype continued to worsen after P14 (*Figure 6—figure supplement 1D–F*). Either GluN2A-containing or GluN2B-containing NMDAR is channel competent and GluN2A-containing NMDARs are present in the cortex of GluN2B KO mice, thus it seems possible that NMDAR's role in callosal circuit development may be separable from its channel activity.

## NMDAR regulates callosal circuit development independent of NMDAR channel activity

$Ca^{2+}$ influx through NMDARs is essential for synaptogenesis, experience-dependent synaptic remodeling, and long-lasting changes in synaptic efficacy such as long-term potentiation (LTP) and long-term depression (LTD) (*Collingridge et al., 2004*; *Lau and Zukin, 2007*). However, accumulating evidence shows that there are NMDAR functions independent of its ion-influx (*Kessels et al., 2013*; *Nabavi et al., 2013*), such as a use-dependent tyrosine dephosphorylation of NMDA receptors is independent of ion flux (*Vissel et al., 2001*), and NMDAR-dependent LTD which can be induced independent of $Ca^{2+}$ influx (*Dore et al., 2016*). In models of ischemic stroke, neuronal death caused by overactivation of NMDAR is also independent of $Ca^{2+}$ influx, but dependent on signaling complexes formed by NMDARs, Src kinase, and Panx1 (*Weilinger et al., 2016*). To address whether channel activity of the NMDAR is required in callosal targeting, we systemically injected MK-801, a non-competitive NMDAR antagonist during CC development. MK-801 enters the open NMDAR channel and binds to the 'blocking site' located deep in the pore, blocking $Ca^{2+}$ influx through NMDAR (*Huettner and Bean, 1988*, *Figure 7A*). Based on previous literature, a single dose of MK-801 for acute i.p. administration is up to 1–10 mg/kg (*Foster et al., 1988*; *Mitrovic et al., 1996*); the daily dose of MK-801 for chronic i.p. administration is around 0.3–0.6 mg/kg (*Nilsson et al.,*

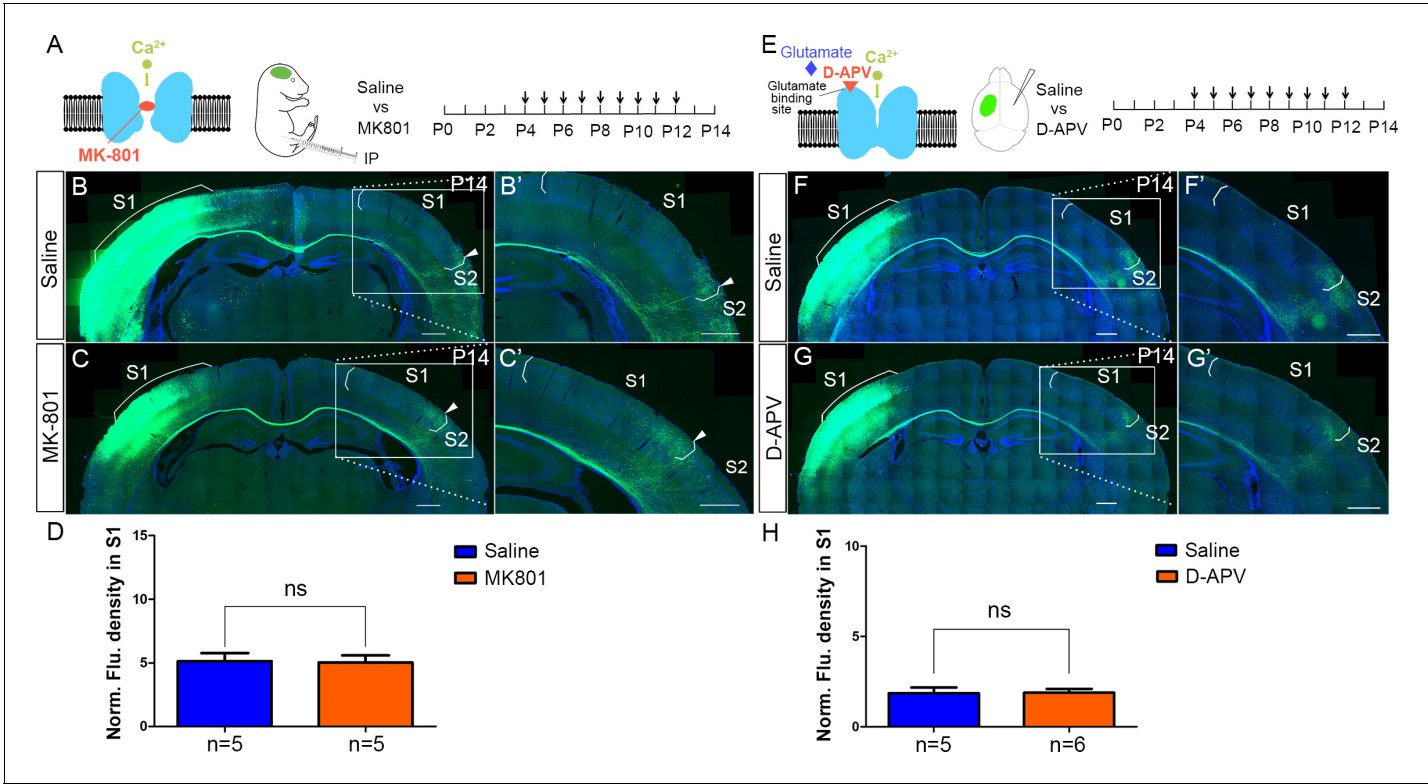

**Figure 7.** NMDAR regulates callosal circuit development independent of NMDAR channel activity. (**A–D**) Blocking $Ca^{2+}$ influx through NMDAR by MK-801. MK-801 enters the open NMDAR channel and binds to the 'blocking site' located deep in the pore (**A**). Callosal innervation patterns in Saline (**B**) and MK-801 (**C**) injected mice at P14. (**D**) Quantification of fluorescence density. p=0.91. (**E–H**) Blocking NMDAR channel opening by D-APV. D-APV competitively inhibits glutamate binding site to NMDAR (**E**). Callosal innervation patterns in saline (**F**) and D-APV (**G**) injected mice at P14. (**H**) Quantification of fluorescence density. p=0.93. Scale bar: 500 µm for all images.

*1997*; *Uttl et al., 2018*; *Zuo et al., 2006*). Previous studies showed that blocking NMDAR with a daily dose of 0.4 mg/kg MK-801 in ferret pups between P14 and P21 disrupted axonal pattern formation by retinal afferents in the lateral geniculate nucleus (LGN) (*Hahm et al., 1999*). We identified a dose of 1 mg/kg MK-801 (see Materials and methods for detail) to block $Ca^{2+}$ influx of NMDAR from P4 to P12 and examined the callosal innervation pattern at P14 (*Figure 7A–D*). Compared with saline control, MK801-treated pups gained weight slowly and developed opisthotonic posturing of limbs and head (data not shown), suggesting the channel function of NMDAR had been blocked. However, the normal callosal innervation pattern in these pups was similar to saline-treated controls (*Figure 7B–D*). Similar weight loss and abnormal behaviors have been reported in neonatal mice after MK-801 administration previously (*Facchinetti et al., 1993*; *Griesbach and Amsel, 1998*; *Wu et al., 2005*). We also performed similar experiments using D(-)−2-amino-5-phosphonopenta-noic acid (D-APV), which competitively blocks the ligand (glutamate) binding site to NMDAR and inhibits channel opening, thereby preventing $Ca^{2+}$ influx (*Morris, 1989*, *Figure 7E*). In a previous study, a one-time infusion of 5 µg D-APV into the basolateral amygdala of adult rat (~250 g) caused memory deficit persistent for at least 4 weeks (*Milton et al., 2008*). Also, blocking NMDARs by intra-cerebral infusion of 0.05 µg D-APV into P7 mouse pup reduced somatic calcium transients in pyrami-dal cells evoked by lateral olfactory tract stimulation, and caused memory deficits both in short-term (3 hr) and long-term (24 hr) odor preference memory (*Mukherjee and Yuan, 2016*). We injected D-APV (5 µg/µl, 0.8–1 µl/injection, see Materials and Methods for detail) into the lateral ventricle of the contralateral S1 twice-daily from P4 to P12 and examined the callosal innervation pattern at P14 (*Figure 7E–H*). Compared with saline control, D-APV-treated pups developed unilateral muscle con-tractions in limbs on the opposite side of the injection (data not shown), suggesting the channel function of NMDAR had been blocked. However, the callosal innervation pattern in these pups was

again similar to saline controls (*Figure 7F–H*). Taken together, these results indicate that NMDAR function in callosal targeting is independent of its channel activity.

## EPHB2 protein expression is decreased in GluN1 KO mice

Our studies to this point led us to consider whether the NMDAR may mediate callosal axon targeting via interaction with other guidance signaling systems. Previous studies have shown that NMDAR subunits bind directly to EPHB axon guidance receptors (*Dalva et al., 2000*). EPHB receptor tyrosine kinases and their transmembrane-ligands, the EPHRIN-B family, mediate short-distance cell–cell signaling and thus regulate many developmental processes at the interface between pattern formation and morphogenesis, including ordered neural maps (*Kania and Klein, 2016*; *Niethamer and Bush, 2019*). Further, several members of the EPH/EPHRIN family, including EPHRIN-B1 and EPHB2, are involved in earlier stages of CC midline axon crossing, strengthening their relevance in this context (*Bush and Soriano, 2009*; *Orioli et al., 1996*; *Robichaux et al., 2016*). EPHB2 and NMDARs colocalize at postsynaptic dendrites, and the extracellular domain of NMDAR interacts directly with EPHB2, an interaction driven by activation of EPHB2 by clustered EPHRIN-B1 expressed in presynaptic axon terminals (*Dalva et al., 2000*; *Nolt et al., 2011*; *Palmer and Klein, 2003*). Mice lacking EPHB2 have reduced levels of NMDARs at synapses in the hippocampus and cortex (*Henkemeyer et al., 2003*; *Sheffler-Collins and Dalva, 2012*), suggesting coordinated localization. EPHB2 also preferentially decreases $Ca^{2+}$-dependent inactivation of GluN2B-containing NMDARs but not GluN2A-containing NMDARs at synapses of mature neurons (*Nolt et al., 2011*). In addition, EPHB2 signaling leads to phosphorylation of GluN2B at tyrosine residue 1472 preventing clathrin-dependent endocytosis, and increasing the surface retention of GluN2B-containing NMDARs (*Chen and Roche, 2007*; *Nolt et al., 2011*; *Takasu et al., 2002*). Taken together, these pieces of evidence suggested us that NMDAR may cooperate with EPHRIN-B/EPHB signaling during initial circuit formation.

We thus examined expression of EPHB2 in NMDAR KO mice by immunostaining. At P5, EPHB2 was found in both cortex and CC of controls (*Figure 8—figure supplement 1A*). However, in GluN1 KO mice, EPHB2 expression in the cortex was reduced (*Figure 8—figure supplement 1B*). Western blots also confirmed that protein levels of EPHB2 were reduced at least 30% in GluN1 KO mice (*Figure 8—figure supplement 1C–E*) while mRNA levels of EPHB2 were unchanged (*Figure 8—figure supplement 1F*). As discussed above, in *Emx1^cre/+; Grin1^fl/fl* mice, GluN1 was selectively deleted in excitatory neurons. Based on previous studies, the expression of GluN1 in interneurons (*Korotkova et al., 2010*), oligodendrocytes, and oligodendrocyte precursor cells (*Káradóttir et al., 2005*) in the cortex should not be affected in *Emx1^cre/+; Grin1^fl/fl* mice. To examine the expression of EPHB2 specifically in GluN1 KO cells, we crossed *Emx1^cre/+; Grin1^fl/fl* mice with Cre-reporter mice – *Rosa26^fs-tdTomato* mice that exhibit tdTomato expression after Cre-mediated recombination. Supporting our hypothesis that loss of GluN1 leads to loss of EPHB2, the punctate staining of EPHB2 on cell membranes was completely lost in red (recombined) cells of *Emx1^cre/+; Grin1^fl/fl; Rosa26^fs-tdTomato* brain sections but not red cells of *Emx1^cre/+; Grin1^wt/wt; Rosa26^fs-tdTomato* brain sections (*Figure 8—figure supplement 2*). Taken together, we demonstrated that the loss of NMDAR caused the loss of EPHB2 selectively on cells that lack GluN1 after excision, thus explaining the 30% reduction in EPHB2 protein expression. Given the known physical association between NMDAR and EPHB2, these data suggest reciprocity in this stabilizing interaction and dendritic localization.

## NMDAR cooperates with EPHRIN-B/EPHB in controlling axon targeting in S1

EPHB2 and NMDARs colocalize at postsynaptic dendrites, and the extracellular domain of NMDAR interacts directly with EPHB2, an interaction driven by activation of EPHB2 by clustered EPHRIN-B1 expressed in presynaptic axon terminals (*Dalva et al., 2000*). This is consistent with the possibility that EPHRIN-B1, expressed by the projecting neuronal axons, signals through EPHB2 and NMDAR, located on the target neurons, to regulate axon extension in the contralateral cortex (*Figure 8A*). To test this prospect, we deleted EPHRIN-B1 in projecting neurons by electroporating vectors of Cre and EGFP at E15.5 in *Efnb1^fl/fl* mice and examined the initial callosal targeting at P6 (*Figure 8B and C*). In projecting neurons lacking EPHRIN-B1, callosal axons extended into the cortex past layer V, and many axons terminated in layer II/III (arrows in *Figure 8C*), similar to that observed in GluN1 KO mice at P6 (*Figure 2B*).

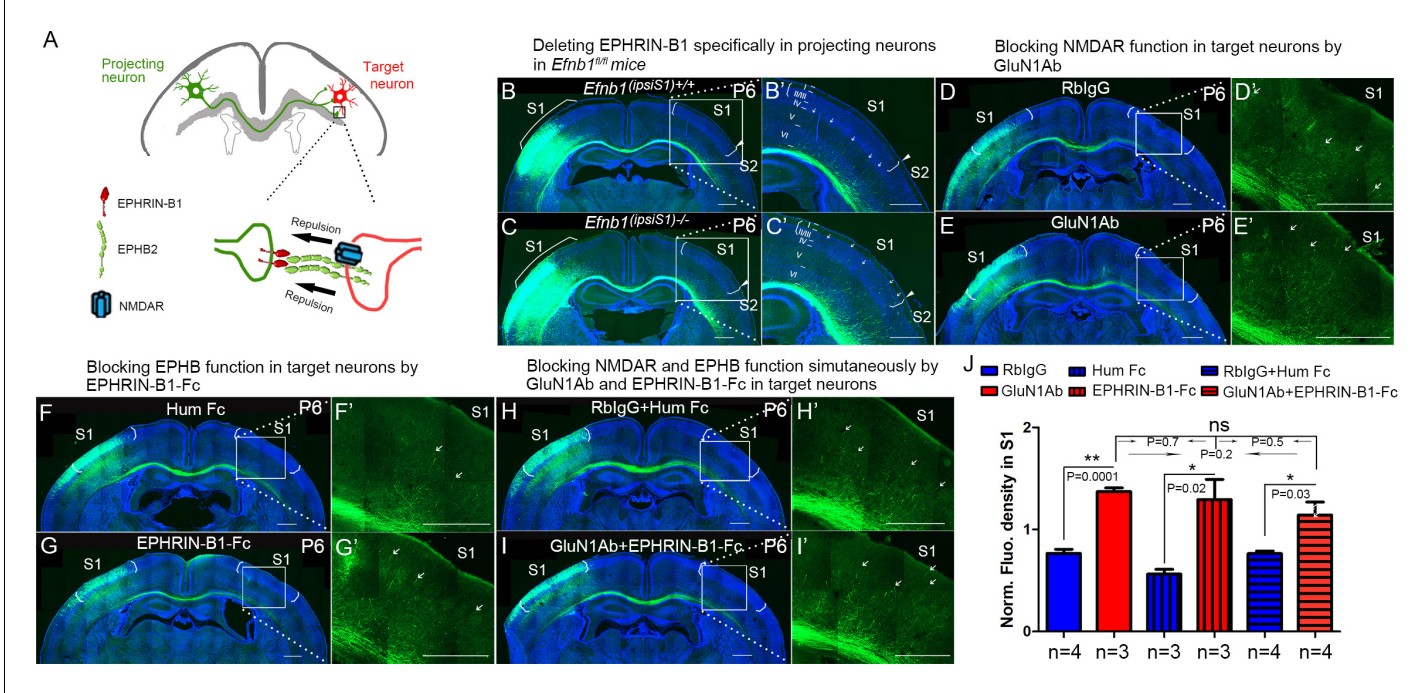

**Figure 8.** NMDARs cooperate with EPHRIN-B/EPHB in controlling axon targeting in S1. (**A**) EPHRIN-B1, expressed by the projecting neuronal axons, signals through EPHB2 and NMDAR, located on the target neurons, regulates axon extension in contralateral cortex. (**B, C**) Deleting EPHRIN-B1 in projecting neurons in *Efnb1*$^{fl/fl}$ mice. Vectors expressing Cre and EGFP were delivered into S1 of all pups from *Efnb1*$^{fl/wt}$ × *Efnb1*$^{fl/wt}$ crosses by in utero electroporation at E15.5. Compared with control mice (**B**), ipsilaterally deleted mice showed earlier callosal innervation at P6 (**C**). (**D, E**) Blocking NMDAR function in target neurons by intraventricular injection of GluN1Ab in contralateral cortex, from P3 to P6. Compared with control RbIgG injected mice (**D**), GluN1Ab injected mice showed earlier callosal innervation at P6 (**E**). (**F, G**) Blocking EPHB function in target neurons by intraventricular injection of EPHRIN-B1-Fc in contralateral cortex, from P3 to P6. Compared with control Hum Fc injected mice, EPHRIN-B1-Fc injected mice showed earlier callosal innervation at P6 (**G**). (**H, I**) Blocking NMDAR and EPHB function simultaneously by GluN1Ab and EPHRIN-B1-Fc in contralateral cortex, from P3 to P6. Compared with control RbIgG + Hum IgG injected mice (**H**), GluN1Ab + EPHRIN-B1-Fc injected mice showed earlier callosal innervation at P6 (**I**). (**J**) Quantification of fluorescence density. Scale bar: 500 μm for all images. Arrows pointed out axon terminals in the target cortex.

The online version of this article includes the following figure supplement(s) for figure 8:

**Figure supplement 1.** The protein but not RNA level of EPHB2 was reduced in *Emx1*$^{cre/+}$; *Grin1*$^{fl/fl}$ mice at P8.

**Figure supplement 2.** The cell membrane expression of EPHB2 was absent in *Emx1*$^{cre/+}$; *Grin1*$^{fl/fl}$; *Rosa26*$^{fs-tdTomato}$ positive cells.

Since this manipulation does not take into account potential compensation by other EPHRIN-B ligands, we blocked all EPHB2 signaling in the contralateral cortex by intraventricular injection of soluble EPHRIN-B1-Fc, from P3 to P6 and examined the initial callosal targeting at P6 (*Figure 8F and G*). EPHRINs have to be clustered in a cell membrane or artificially with, for example, antibodies to efficiently activate EPH receptors (*Davis et al., 1994*). Unclustered soluble EPHRINs bind EPH receptors but act as antagonists (*Vearing and Lackmann, 2005*). Compared with littermate controls, contralateral injection of soluble EPHRIN-B1-Fc led to increased callosal innervation at P6 with axons penetrating to superficial layers (*Figure 8F, G, and J*). These results support the idea that EPHRIN-B1-EPHB2 play important roles in controlling callosal axon penetration into the contralateral S1 cortex.

To further determine whether there is collaboration between NMDAR and EPH-B signaling, we developed a way to assess blockade of both systems and determine if this blockade was additive, synergistic or likely acting in the same pathway. We blocked NMDAR in contralateral/target cortex by intraventricular injection of anti-GluN1 antibodies, from P3 to P6, and examined the initial callosal targeting at P6 (*Figure 8D and E*). Compared with controls, contralateral injection of anti-GluN1 antibodies caused increased callosal innervation at P6 and penetration to superficial layers of cortex (*Figure 8D, E, and J*). Critically, blocking NMDAR and EPHRIN-B1-EPHB2 signaling simultaneously in contralateral cortex by injecting both anti-GluN1 antibody and EPHRIN-B1-Fc did not show any

additive or synergistic effects (*Figure 8H–J*). This suggests that in fact NMDAR and EPHRIN-B1-EPHB2 are in the same pathway in regulating callosal targeting in S1 and is consistent with our other findings.

## Discussion

In this study, we found that NMDARs cooperate with EPHRIN-B/EPHB in controlling callosal circuit formation and demonstrated that it is GluN2B-containing NMDARs in target S1 cortex that regulate callosal axon targeting in S1. In mutants where either GluN2B or GluN1 are disrupted, there is excess axonal growth throughout S1. Either genetic or antibody-mediated loss of NMDAR specifically in the target hemisphere disrupted this projection. We found that this begins at P6 when axons that should terminate in deep cortical layers of target S1 instead project more superficially. Once this targeting defect begins, it continues to worsen. We believe this phenotype is most consistent with a loss of a repellant activity that normally restricts commissural axonal projections to the S1 cortex at initial stages of this process (*Figure 2*). Once these axons aberrantly penetrate through the deeper cortical layers of S1 in the GluN1 mutants, this phenotype continues to worsen. A systematic role for NMDAR in this process was a surprise and has significant implications for disorders where NMDAR function is lost due to mutations or due to antibody-mediated disruption of NMDAR surface distribution. We predict this would be a potent disruptor of circuit formation during development certainly in this callosal circuit, but likely others as well.

### Mechanisms for the formation of homotopic callosal projection during development

In previous studies (*Zhou et al., 2013*), we showed that the medial-lateral topography of callosal neurons is tightly constrained by the D-V position of their axons within the CC. The axon position within the CC determines the contralateral cortical projection, with more dorsally located axons projecting medially and ventrally located axons projecting laterally. The complementary expression of chemotropic guidance cue Semaphorin3A (Sem3A) and its receptor Neuropilin-1 (Nrp1) contributes to this axonal order within the CC. The studies of genetic manipulations showed that Sema3A/Nrp1 signaling contributes to the topographic order of axons within the CC but is not involved in the axon position-dependent projection to the contralateral target cortex. Here, in this study, we demonstrated that contralateral/target expression of NMDAR controls the pattern of callosal projections to contralateral target S1. Interestingly, despite the disruption of targeting within S1 in mice with disrupted NMDAR, there is little evidence of ectopic projection to other cortical areas. This indicates that there are still other cues that regulate the generic projection to S1 but that the mechanisms we have uncovered help govern the distribution of these axons within S1.

### NMDARs cooperate with EPHRIN-B/EPHB to regulate axon extension into the cortex

Given the longstanding understanding of important roles for neural activity in the wiring of circuits and the generation of cortical maps, we expected that the function of NMDAR in this process would be due to the critical role of NMDAR in excitatory activity in the developing brain as synapses form. However, our data show that NMDAR ion channel function is not necessary for its role in somatosensory callosal targeting. These data are consistent with the idea that NMDAR protein complexes containing GluN1 and GluN2B are critical for commissural projection. Our findings suggest that without NMDAR containing GluN1 and GluN2B in the target cortex, there is earlier and increased callosal innervation of S1 starting at P6, when callosal axons start entering the contralateral cortex. We believe that this observation indicates that either loss of NMDAR leads to an increase in the attractiveness of S1 cortex or that loss of NMDAR leads to loss of a repellant activity in the cortex that normally limits axon growth into superficial layers of S1, until axons reach the S1/S2 border. This realization led us to consider whether there might be a role for NMDAR complexes in collaborating with already known axon guidance mechanisms.

EPHRIN-B/EPHB signals are well-known repulsive axon guidance cues. Because EPHB2 is necessary for localization of NMDAR to postsynaptic terminals, we wondered if this requirement was reciprocal and found indeed that it is – when NMDARs are lost, membrane based EPHB2 protein expression in target neurons is lost as well (*Figure 8—figure supplement 2*). This result suggests

that NMDAR reciprocally regulates the localization of EPHB2. In addition, the NMDAR interaction with EPHB2 is driven by the stimulation of EPHB2 by clustered EPHRIN-B1 expressed in presynaptic axon terminals, indicating that the interaction may regulate axon guidance and termination. When we deleted EPHRIN-B1 only in projecting callosal axons, this also led to excess ingrowth into the S1 cortex visible from P6 to P8. That this phenotype is less severe than the NMDAR phenotype is most likely due to redundancy with other EPHRIN-B ligands also expressed in projecting axons. To globally disrupt EPHB-EPHRIN-B interactions we injected EPHRIN-B1-Fc fragments and found that these also disrupted the somatosensory callosal targeting as efficiently as anti-NMDAR antibodies; furthermore, these two treatments were neither additive nor synergistic, implying they act through a common pathway. Further studies will examine whether this phenotype is due to loss of repulsive activity of EPHRIN-B/EPHB signals, and whether it is through forward EPHB or reverse EPHRIN-B signaling or both.

### The refinement of callosal innervation pattern

The production of transient, exuberant axons, and axonal branches is a general phenomenon in development across species and systems (*De León Reyes et al., 2019*; *Fenlon et al., 2017*; *Hand et al., 2015*; *Innocenti, 2020*; *Innocenti and Price, 2005*; *Luo and O'Leary, 2005*; *Ribeiro Gomes et al., 2020*). Our data indicate that NMDAR are required to collaborate with EPHRIN-B/EPHB in target cortex, and the most parsimonious conclusion is that this acts as a repellant to limit early growth of callosal axons into S1 cortex. These phenotypes are seen first at P6 when axons inappropriately enter superficial cortex in S1 so that by P30 (when the animals die) there is a nearly sixfold increase in axonal extension into S1. This is consistent with a role for NMDAR in controlling initial projection into S1 but probably also later regulating refinement of these inappropriate projections by pruning – both core functions of axonal repellants. Previous studies have observed the role of NMDAR in axon elimination (*Colonnese and Constantine-Paton, 2006*; *Henson et al., 2017*; *Personius et al., 2016*; *Rabacchi et al., 1992*; *Rajan et al., 1999*; *Zhang et al., 2013*). In vivo time-lapse images of retinal axons from albino *Xenopus* tadpoles show that correlated neural activity influences axon branch dynamics of retinal axons in the optic tectum (*Ruthazer et al., 2003*). The axon branches whose activity is not correlated with that of their neighbors are selectively eliminated. However, the selective elimination can be abolished by blocking the $Ca^{2+}$ influx of NMDAR by its antagonist MK-801. When we blocked the $Ca^{2+}$ influx of NMDAR by MK-801 or D-APV, even though the overall callosal innervation pattern of somatosensory cortex was not affected, there were some subtle, local laminar-specific innervation differences at the S1/S2 border, suggesting that the refinement of axon arborization may rely on the $Ca^{2+}$ influx-mediated activity changes through NMDAR.

Regarding previous roles of EPHRIN-B/EPHB signaling in pruning, this pathway was previously shown to be involved in infrapyramidal bundle pruning in dentate gyrus (*Xu and Henkemeyer, 2009*) and it is known that activation of EPHB by EPHRIN-B potentiates $Ca^{2+}$ influx of NMDA receptor (*Takasu et al., 2002*). These lines of evidence suggest that NMDAR and EPHRIN-B/EPHB signaling may cooperate in the refinement of the callosal innervation pattern and perhaps in other circuits. Further studies will examine how and whether NMDAR regulates EPHRIN-B/EPHB signaling directly and the detailed molecular mechanism underlying their collaboration to regulate callosal innervation patterns in the primary somatosensory cortex.

## Materials and methods

### Experimental model and subject details

All animal protocols were in accordance with the regulations of the National Institute of Health and approved by the University of California San Francisco Institutional Animal Care and Use Committee (IACUC). Floxed GluN1*Grin1* allele (Stock #005246), EMX1-Cre (Stock #005628) and *Rosa26<fs-tdTomato>* Cre reporter allele (Stock # 007914) were obtained from Jackson Laboratories (Bar Harbor, ME, USA). Floxed *Grin2a* and *Grin2b* alleles were provided by the laboratory of Prof. Roger Nicoll. Floxed *EphrinB1* allele was provided by the laboratory of Prof. Jeffrey Bush. Wild-type CD1 mice were obtained from Charles River Laboratories. Male and female embryos at embryonic (E) 12.5 and E15.5 were used for the in utero electroporation, and pups between postnatal day 0 (P0) to 30 (P30) for the experiments.

## In utero electroporation

DNA solution including the plasmid and 0.04% fast green was injected into the medial region of the lateral ventricle of the embryonic brain with a glass micropipette. Electrical pulses then were delivered to embryos by electrodes connected to a square-pulse generator (ECM830, BTX). For each electroporation, five 35 V pulses of 50 ms were applied at 1 s intervals. After the electroporation, the uterus was returned to the abdominal cavity, followed by suturing of the abdominal wall and skin. Mice were perfused at different postnatal stages using 4% paraformaldehyde followed by post-fixed overnight and incubation in 30% sucrose at 4°C. 35 μm-thick coronal sections were obtained using cryostat sectioning.

## Plasmid

Plasmid of pCAGGS-Cre and pCAGGS-CreERT was obtained from Addgene. The ubiquitin-EGFP plasmid used was from a previous study (*Zhou et al., 2013*).

## Antibodies: Antibodies for intraventricular injection

Commercial anti-NMDAR antibody is against amino acid residues 385–399 in the N-terminus of GluN1 and was made in Rabbit (AGC-001, Alomone labs). Rabbit IgG (#31235, Invitrogen) served as control. Recombinant Mouse EPHRIN-B1 Fc Chimera Protein was obtained from R and D (473-EB). Recombinant Human IgG1 Fc Protein (110-HG, R and D) served as control. Antibodies for immunostaining: Rabbit anti-GluN1 (1:500, AGC-001, Alomone labs), anti-vGlut2 (1:200, AB2251, Millipore), goat anti-EphB2 (1:50, AF467, R and D), anti-cleaved caspase-3 (#9661S, Cell Signaling), anti-Rabbit 594 (#711-585-152, Jackson ImmunoResearch), and anti-guinea pig 488 (A-11073, Invitrogen). Antibodies for western blot: rabbit anti-GluN1 (AB9864, Millipore), rabbit anti-GluN2A (#4205, Cell signaling), rabbit anti-GluN2B (#4212, Cell signaling), goat anti-EPHB2 (1:1000, AF467, R and D), rabbit anti-GAPDH (1:5000, #2118, Cell Signaling), rabbit anti-β tubulin (1:5000, #2128, Cell Signaling), IRDye 680RD Donkey anti-Goat IgG(H+L) Ab (1:10,000, #926–68074, Li-Cor), and IRDye 800CW Goat × Rabbit IgG(H+L) Ab (1:20,000, #925–32211, Li-Cor).

## Intraventricular injection

Antibodies/Fc-fragment was injected to lateral ventricular of pups by glass pipette with a sharp bevel at 45° (BV-10 Micropipette Beveler, Sutter instrument). The diameter of pipette tip was ~40–80 μm (*Vogt et al., 2015*). The concentrations for antibody injections were 3.2 μg/μl for the commercial anti-GluN1 antibody and Rabbit IgG. The concentrations for Fc injections were 2 μg/μl for EPHRIN-B1 Fc and Human IgG1 Fc. Antibodies/Fc-fragment was injected twice-daily and the injection volume was 0.8–1 μl for each injection.

## MK-801 systemic injection

EGFP positive pups were given intraperitoneal (i.p.) injection of MK-801 (1 mg/kg, M107-25MG, Sigma) or equivalent volume of 0.9% saline once-daily. See the following for the criteria of selecting effective dose of MK-801. Based on literature, the single dose of MK-801 for acute i.p. administration is up to 1–10 mg/kg (*Foster et al., 1988*); the daily dose of MK-801 for chronic i.p. administration is around 0.3–0.6 mg/kg (*Nilsson et al., 1997*; *Uttl et al., 2018*; *Zuo et al., 2006*). To optimize the dose for chronic administration, MK-801 was given to three groups of pups (each group had four pups with mixed genders) from P4 to P9 with the doses of 1 mg/kg, 10 mg/kg, and 20 mg/kg, respectively. Saline was given to four pups of the same litter as control. We measured body weights of all pups every day before i.p. administration. The pups in saline group always had abundant milk in their stomachs and gained weight rapidly. The pups with the dose of 1 mg/kg had milk in their stomachs and gained weight, but very slowly. Some even lost weights. All MK-801-treated pups developed abnormal behavior-opisthotonic posturing for heads and four limbs, similar to the abnormal postures in children with anti-NMDAR encephalitis (*Florance et al., 2009*). The doses of 10 mg/kg and 20 mg/kg were too close to lethal dose and the pups lost weight and died within 2–3 days. Thus, we chose 1 mg/kg for our experiment. The weight loss and abnormal behavior had been reported in neonatal mice after MK-801 administration (*Facchinetti et al., 1993*; *Griesbach and Amsel, 1998*; *Wu et al., 2005*).

## D(-)−2-amino-5-phosphonopentanoic acid (D-APV) intraventricular injection

Because D-APV poorly penetrates the blood–brain barrier (BBB) when administered systemically, we injected it directly into the lateral ventricle of targeted cortex. EGFP positive pups were given D-APV (5 µg/µl, Millipore-sigma, 165304–5 MG) twice-daily at an injection volume of 0.8–1 µl/injection. See section Intraventricular injection for injection details. See the following for the criteria of selecting effective dose of D-APV. Based on literature, one-time infusion of 5 µg D-APV into the basolateral amygdala of adult rat (~250 g) caused memory deficit persistent for at least 4 weeks (*Milton et al., 2008*). Blocking NMDARs by intracerebral infusion of 0.5 µl of 500 µM D-APV into P7 mouse pup reduced somatic calcium transients in pyramidal cells evoked by lateral olfactory tract stimulation, and caused memory deficits both in short-term (3 hr) and long-term (24 hr) odor preference memory (*Mukherjee and Yuan, 2016*). 0.5 µl of 500 µM D-APV is equal to 0.05 µg D-APV (molecular weight of D-APV: 197.13). To optimize the dose for chronic administration, intraventricular D-APV injection was given to two groups of pups (each group had four pups with mixed genders) from P4 (~4 g) to P9 (~8 g) with the concentrations of 5 µg/µl and 10 µg/µl twice-daily (0.8–1 µl/injection), respectively. Saline was given to four pups of the same litter as control. The body weight of all pups was measured before intraventricular injection. There were no body weight differences between saline-treated and D-APV-treated mice. However, D-APV-treated pups in both 5 µg/µl-treated and 10 µg/µl-treated groups developed unilateral muscle contractions in limbs on the opposite side of the injection within 10 min after injection, similar to the abnormal movements seen in MK-801-treated mice but only restricting in one side of body. No pups died during the 6 days of injections. As 5 µg/µl-treated and 10 µg/µl-treated had no dose-dependent effect on the abnormal behavior and based on dosage of D-APV used in literature, we chose 5 µg/µl D-APV treatment for our experiment.

## Slice preparation and imaging

Mice were perfused with saline followed by 4% paraformaldehyde in phosphate buffered saline (PBS), pH 7.4. Brains were removed from mice and post-fixed in 4% paraformaldehyde overnight before being placed in 30% sucrose solution. The brains were then cut into 35 µm sections with cryostat (Leica VT1200S). Sections were imaged by Zeiss Axioscan Z.1 (Zeiss, Thornwood, NY, USA) with a 20× objective.

## Immunostaining

Mouse pups were perfused with saline followed by 4% paraformaldehyde in PBS, pH 7.4. Brains were removed from mice and post-fixed in 4% paraformaldehyde overnight before being placed in 30% sucrose solution. The brains were then cut into 12 µm sections with cryostat (Leica VT1200S). Non-specific binding was blocked by adding 5% normal goat/donkey serum during pre-incubation and incubations in 1× PBS containing 0.05% TritonX-100. The primary antibodies were applied overnight at 4°. Secondary antibodies were applied for 1–2 hr at 4° and nuclei were stained with DAPI. Slides were mounted with Prolong Gold Anti-fade Mountant (P36930, Invitrogen).

## Western blotting

Mouse brain tissue of somatosensory cortex from *Emx1*$^{cre/+}$; *Grin1*$^{fl/fl}$ mice or littermate controls at P8 were collected for western blotting. Five mouse samples for each group. The protocol we used as described before (*Yabut et al., 2015*).

## Primers for RT-PCR

Primers for EphB2: Forward Primer-ATTATTTGCCCCAAAGTGGACTC; Reverse Primer-GCAGCGGGGTATTCTCCTTC.

## RT-PCR

Mouse brain tissue of somatosensory cortex from *Emx1*$^{cre/+}$; *Grin1*$^{fl/fl}$ mice or littermate controls at P8 were collected for RT-PCR. Six mouse samples for each group. The protocol we used as described before (*Yabut et al., 2015*).

## Quantification and statistical analysis

### Callosal axon distribution analysis

Using the segmented line tool in ImageJ, a line was drawn with a width of 200 pixels from medial S1 to lateral S1/S2 border along cortical layer II/III. Fluorescence distribution was measured along the line by using 'Plot Profile' under 'Analyze' in ImageJ and produced data sets with distance points along the line (X) and fluorescence intensity (Y). The data was exported to Excel. In Excel, fluorescence intensity values (Y) were normalized by dividing by the max Y fluorescence value for that group. Finally, results were analyzed by using XY statistics in Prism version 5.0 (GraphPad Software).

### Callosal axon density analysis

Sections were imaged using a Zeiss Axioscan Z.1 (Zeiss, Thornwood, NY, USA) with 20× objective over whole brain section. Each image was made up by the compression of three slices in 4 μm Z-stack. For each brain, only one section was chosen for data quantification. The callosal axon density (fluorescence density) in S1 was quantitatively analyzed by ImageJ software. First, each picture was converted to an 8-bit image, and then Brightness/Contrast (*Figure 2—figure supplement 3A*) and Threshold Ranges (*Figure 2—figure supplement 3B*) were set. Threshold Range was set to eliminate background fluorescence from affecting fluorescent density. Second, the cortical S1 region in the target side was identified according to Dapi staining as previously described (*Zhou et al., 2013*) and a box was drawn to encompass only the S1 (box I in *Figure 2—figure supplement 3B*). Third, fluorescence density was quantified in the S1 by counting the number of pixels within the threshold range and dividing by total number of pixels in the area. This is done by selecting 'Area Fraction' and 'Limit to Threshold' in ImageJ → Analyze → Set Measurement. Finally, the axon density in S1 was normalized by the average fluorescence density of midline for each image (box II in *Figure 2—figure supplement 3B*). The average fluorescence density of midline was calculated by measuring the fluorescence density of six non-overlapping points around the midline and averaging the values. A fixed sized box was used for all measurements of midline fluorescence density. Since a fixed sized box was used (total number of pixels is fixed for all), 'Area' instead of 'Area Fraction' was used for analysis. Results were analyzed by using two-tailed t-test in Prism version 5.0 (GraphPad Software). Please note that control groups and experimental groups followed exactly the same settings of measurements.

## Acknowledgements

We thank Prof. Roger Nicoll and Jillian Iafrati for providing *Grin2a*$^{fl/fl}$ and *Grin2b*$^{fl/fl}$ mice. We thank Wucheng Tao in Prof. Roger Nicoll's lab for suggestions on the experimental design of MK-801 in vivo injection. We thank Drs. Ariele L Greenfield, Christopher M Bartley, and Michael R Wilson for critical comments and editing of this manuscript. Work in the Pleasure lab was supported by R56 MH119435, R01MH122471, UCSF Weill Institute for Neurosciences Innovation Award, and UCSF Marcus Program in Precision Medicine Innovation Transformative Integrated Research Initiative. JOB was supported by R01DE023337 from NIH/NIDCR.

## Additional information

### Funding

| Funder | Grant reference number | Author |
| --- | --- | --- |
| National Institute of Mental Health | R01MH119435 | Jing Zhou<br>Yong Lin<br>Trung Huynh<br>Hirofumi Noguchi<br>Samuel J Pleasure |
| National Institute of Mental Health | R01MH122471 | Jing Zhou<br>Yong Lin<br>Trung Huynh<br>Hirofumi Noguchi<br>Samuel J Pleasure |

| National Institute of Dental and Craniofacial Research | R01DE023337 | Jeffrey O Bush |
|---|---|---|
| University of California, San Francisco | UCSF Weill Institute for Neurosciences Innovation Award | Jing Zhou<br>Yong Lin<br>Trung Huynh<br>Hirofumi Noguchi<br>Samuel J Pleasure |
| University of California, San Francisco | UCSF Marcus Program in Precision Medicine Innovation Transformative Integrated Research Initiative | Jing Zhou<br>Yong Lin<br>Trung Huynh<br>Hirofumi Noguchi<br>Samuel J Pleasure |

The funders had no role in study design, data collection and interpretation, or the decision to submit the work for publication.

## Author contributions

Jing Zhou, Conceptualization, Data curation, Formal analysis, Validation, Investigation, Visualization, Methodology, Writing - original draft, Writing - review and editing; Yong Lin, Data curation, Formal analysis, Writing - review and editing; Trung Huynh, Data curation, Investigation, Writing - review and editing; Hirofumi Noguchi, Data curation, Formal analysis, Investigation, Writing - review and editing; Jeffrey O Bush, Resources, Methodology, Writing - review and editing; Samuel J Pleasure, Conceptualization, Resources, Supervision, Funding acquisition, Investigation, Methodology, Writing - original draft, Project administration, Writing - review and editing

## Author ORCIDs

Jing Zhou https://orcid.org/0000-0003-2809-7097
Hirofumi Noguchi http://orcid.org/0000-0002-9779-4956
Jeffrey O Bush https://orcid.org/0000-0002-6053-8756
Samuel J Pleasure https://orcid.org/0000-0001-8599-1613

## Ethics

Animal experimentation: This study was performed in strict accordance with the recommendations in the Guide for the Care and Use of Laboratory Animals of the National Institutes of Health. All of the animals were handled according to approved institutional animal care and use committee (IACUC) protocols (AN176415) of the University of California San Francisco.

## Decision letter and Author response

Decision letter https://doi.org/10.7554/eLife.59612.sa1
Author response https://doi.org/10.7554/eLife.59612.sa2

# Additional files

## Supplementary files

• Transparent reporting form

## Data availability

All data generated or analyses during this study are included in the manuscript.

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
