## [Decision Letter]

**Acceptance summary:**

Your paper highlights a novel mechanism for the development of callosal projections from primary somatosensory cortex (S1), through demonstrating that the normal termination pattern of callosal projections are disrupted in cortex-specific NMDAR mutants. Rather than projecting selectively to the primary/secondary somatosensory cortex (S1/S2) border, axons are uniformly distributed throughout S1, with this pattern intensifying until a premature death. You also suggest that GluN2B-containing NMDA receptors mediate this phenotype during callosal innervation of somatosensory cortex and is independent of ion channel function. Finally, you propose that NMDAR functions with the ephrinB/EphB system of guidance molecules.

**Decision letter after peer review:**

Thank you for submitting your article "NMDA receptors control cortical axonal projections via EPHRIN-B/EPHB signaling" for consideration by *eLife*. Your article has been reviewed by three peer reviewers, and the evaluation has been overseen by a Reviewing Editor and Jonathan Cooper as the Senior Editor. The reviewers have opted to remain anonymous.

The reviewers have discussed the reviews with one another and the Reviewing Editor has drafted this decision to help you prepare a revised submission.

Summary:

Your study points to a novel mechanism for the development of callosal projections from primary somatosensory cortex (S1), namely, that NR2B-containing NMDA receptors are required to mediate the normal termination pattern of callosal projections. You also show that NR2B-containing NMDAR during callosal innervation of somatosensory cortex is independent of ion channel function in these receptors. An additional, novel slant is your proposal that NMDARs act as accessory axon guidance molecules via Eph/ephrin repulsion. You pose a model where these well-studied axon guidance molecules interact directly with the NMDAR, to coordinate the formation of axonal projections, again, indicating non-ionotrophic functions of the NMDAR.

From the appended reviews, you will see that the reviewers thought that the claims in the manuscript for the NR2B-containing NMDAR function during callosal innervation are well supported. However, all three reviewers questioned the link between the ephrin/Eph system and NMDAR function, and considered the analyses related to this link preliminary. They thought that this aspect should be strengthened, either through additional experiments (in vitro tests of repellent activity) or by toning down the discussion and potentially removing the link to ephrinB/EphB signaling from the title and Abstract. While it may not be possible to do the additional suggested experiments since lab shutdown and restarting, the revisions on data presentation and textually in the discussion are feasible and could be done in the coming months.

For the glutamate receptor data:

1) In the GluN2A and GluN2B floxed mice: at the ages studied, it is unclear how much GluN2A is expressed.

2) With MK801 and APV application to block the NMDAR, how effective are these drugs at blocking the NMDAR and what is the area of effect of the drugs? Are other activity-dependent events blocked in these animals (e.g., LGN lamination, whisker barrelette formation in PrV)?

The drugs blocking NMDAR activity lack sufficient controls for the conclusions stated in the manuscript: Loss of the innervation pattern of innervation is variable and the S1/S2 column indicates a possible dependency on activity, leading to other possible interpretations. Changes in the overall cortical activity may affect the stabilization of callosal projections, and therefore acute pharmacological blocking of NMDAR may not mimic the KO or antibody treatments. Early and permanent loss of NMDAR activity can trigger changes in the intrinsic excitability of the neurons that could override the phenotypes of increased innervation. This could be discussed textually

For the Eph/ephrin data:

3) You argue EphB2 protein levels are reduced in the cortex of conditional NR1 floxed mice and disruption of ephrin-B/EphB signaling in this system leads to similar termination defects as deleting NR2B. This is an apt argument but the reviewers felt that this was only a first step in the required analyses.

Reviewer #3:

4) Figure 8. By Western blot analysis, the reduction seems 30% compared to controls. This is not dramatic and instead suggests other mechanisms at work, since most heterozygous mutants (with target protein reductions at 50% or more) do not show phenotypes.

5) Partial reduction of EphB2 protein raises the possibility that further reductions of EphB2 signaling may enhance the phenotype. Instead, blocking NMDAR plus EphB2 functions (by injecting NR1Ab and ephrin-B1-Fc) did not enhance the phenotype of the single manipulations. One would expect additive effects.

6) Figure 8, only the timepoint P6 is shown. Rev 5 also points out that between P6 and P10 there is more variability in contralateral innervation correlating with the anterior-posterior axis. You should show ephrin B electroporations analyzed at P14-P30, or remove and/or tone down the text.

Reviewer #4:

7) EphB-ephrin-B signaling and NMDAR function likely has to do with EphB-ephrin-B repulsive cues or that EphBs interact and modulate NMDAR function, and regulate excitatory synapse development in forebrain. It seems at least equally likely that in the latter case, EphB modulation of NMDAR may mediate the events described in this manuscript, on synapse development and maintenance, not necessarily on repulsion/guidance.

8) EphB-NMDAR interaction results in phosphorylation of GluN2B, increased synaptic localization of the NMDAR, and increased NMDAR dependent Ca influx. Loss of EphB-ephrin-B signaling might cause a reduction in NMDAR Ca influx, but not a complete block of ionotropic signaling. To demonstrate that the findings in Figures 7 and 8 are linked, the authors should inject NMDAR blockers (after conducting the control experiments in point 2), in the ephrin-B1 knockout mice. The anti-NMDAR antibodies simply remove the receptor from the synapse. This does not address whether the effects are independent of the ionotropic modulation of the channel and should be discussed.

Revisions expected in follow-up work:

1) In implicating repellent activity that would restrict commissural axonal projections to the S1 cortex, the evidence for NMDAR via Eph B2 function as repellents for elongating callosal axons is lacking. in vitro experiments demonstrating repellent activity of EphB downstream of NMDAR would be necessary to make this claim. Given the time required to develop and execute such an assay, and the pandemic delays, you could take as long as needed to do this experiment, if you choose to do so.

2) Reviewers 4 and 5 point out that these pathways could be involved in refinement and pruning of axons: Reviewer 5 raises the possibility that in the mutants, axons are stabilized, to result in extra numbers of callosal neurons. NMDAR could thus be responsible for repulsion during refinement. Reviewer 4's comments on EphBs interacting with excitatory synapse development could play into this explanation. More high-resolution analysis of the impact of your manipulations on axonal branching would address the other reviewers' questions regarding whether the changes see are due defects in branching and/or refinement. This aspect as in the in vitro experiments require additional experiments which could understandably not be possible at this time.

[Editors' note: further revisions were suggested prior to acceptance, as described below.]

Thank you for submitting your article "NMDA receptors control development of somatosensory callosal axonal projections" for consideration by *eLife*. Your article has been reviewed by three peer reviewers, and the evaluation has been overseen by a Reviewing Editor and Jonathan Cooper as the Senior Editor. The reviewers have opted to remain anonymous.

The reviewers have discussed the reviews with one another and the Reviewing Editor has drafted this decision to help you prepare a revised submission.

Summary:

Your revision highlights a novel mechanism for the development of callosal projections from primary somatosensory cortex (S1), through demonstrating that the normal termination pattern of callosal projections are disrupted in cortex-specific NMDAR mutants. Rather than projecting selectively to the primary/secondary somatosensory cortex (S1/S2) border, axons are uniformly distributed throughout S1, with this pattern intensifying until a premature death. You also suggest that GluN2B-containing NMDA receptors mediate this phenotype during callosal innervation of somatosensory cortex and is independent of ion channel function. Finally, you propose that NMDAR functions with the ephrinB/EphB system of guidance molecules.

Due to constraints put in place by COVID, in many cases you have had to reduce the strength of your claims rather than conduct new experiments, and you have satisfactorily downplayed the link between the ephrin/Eph system and NMDAR function. While you were not able to directly show that the NMDAR blockers used were effective at blocking NMDAR channel function, the demonstration of a potential link between activity-independent and activity-dependent signaling systems to impact axon development is considered exciting and novel.

The previous review stated that you could look to performing follow-up work in the future on: (a). in vitro experiments demonstrating repellent activity of EphB downstream of NMDAR to fortify the link between EphB and NMDAR expression levels. (b). experiments to show the effects of inhibiting Eph/ephrin signaling at later time points, especially on examining refinement and pruning of axons. We hope you will be able to pursue these interesting studies when conditions improve.

Revisions:

The rebuttal letter was difficult to follow. It was not clear where in the manuscript some of the changes were made. Figures were not labelled and main figures were mixed with supplementary figures. However, the following points appear not to have been addressed adequately:

1) In response to the criticism of overstating the 30% reduction in EphB2 protein expression in the Grin1 CKOs, the authors have provided images showing EphB2 immunofluorescence in Emx1Cre;Grin1fl/fl;tdTomato mice (Figure 8—figure supplement 2). While the pictures in this new figure suggest loss of EphB2 IF in Tom+;Grin1 cKO cells, but not in Tom+;control cells, the results should be quantified. Presenting images without quantification is not convincing. Please quantify and add the results to the paper.

2) Please address the following points that were raised in the previous critique and appear not to have been answered:

a) show images of all controls and experimental groups of Figure 3 in Supplementary data.

b) Provide better evidence for NR1 protein downregulation in the conditional NR1 knockouts (Figure 1—figure supplement 1).

---

## [Author Response]

Revisions for this paper:For the glutamate receptor data:1) In the GluN2A and GluN2B floxed mice: at the ages studied, it is unclear how much GluN2A is expressed.

We have performed western blot of S1 at P8 and found GluN2A is expressed at P8. We have added “To address previous studies that found that GluN2A was expressed at lower levels during development we checked the expression of GluN2A in S1 at P8, and found that it was dramatically diminished in *Emx1^cre/+^*; *Grin2A^fl/fl^* mice (Figure 6—figure supplement 2A, 2B). While the expression of GluN2B was the same in control and *Emx1^cre/+^*; *Grin2A^fl/fl^* mice (Figure 6—figure supplement 2A, 2C), and similarly GluN2A expression was unchanged in *Emx1^cre/+^*; *Grin2B^fl/fl^* mice (Data not shown)” to the text when mentioned about “GluN2B, but not GluN2A, is required for callosal axon targeting”.

2) With MK801 and APV application to block the NMDAR, how effective are these drugs at blocking the NMDAR and what is the area of effect of the drugs? Are other activity-dependent events blocked in these animals (e.g., LGN lamination, whisker barrelette formation in PrV)?

We agree with these points raised by the reviewer. These were the exactly concerns we had when we began the experiments. We consulted local electrophysiology labs who suggested we use (+)-MK-801 hydrogen maleate and D(-)-2-amino-5-phosphonopentanoic acid (D-APV) which are most the effective isoforms for in vivo studies. We also carefully examined the previous literature to determine the dosage range for chronic treatment studies and tested on our own to find the optimal dosage (see the details in Materials and methods). Based on the literature, weight loss and abnormal behavior have been reported in neonatal mice after MK-801 administration (Facchinetti et al., 1993; Griesbach and Amsel, 1998; Wu et al., 2005) which can be used as a sign of effective MK-801 administration. We also found weight loss and abnormal behaviors after systemic i.p. MK-801 administration in mouse pups. For local D-APV injection, we did not see weight loss for pups but did see abnormal behaviors after injection.

We searched the original literature to address the question “Are other activity-dependent events blocked in these animals (e.g., LGN lamination, whisker barrelette formation in PrV)?” For the LGN, we calculated that the dose used for ferret pups was 0.4mg/kg MK-801 daily. Our dose was 1mg/kg MK-801 daily which is significantly higher than the published dose. It further suggested our dose is effective in mouse pups. So we add this sentence of “Previous studies showed that blocking NMDAR with a daily dose of 0.4mg/kg MK-801 in ferret pups between P14 and P21 disrupted axonal pattern formation by retinal afferents in the lateral geniculate nucleus (LGN) (Hahm et al., 1999)” in the text when mentioned about “NMDAR regulates callosal circuit development independent of NMDAR channel activity”. For PrV (Mitrovic et al., 1996), they used silicone carriers to locally apply MK-801 and D-APV, it was not clear how much MK-801 and APPV were delivered to the targeted tissue.

Of course, the most direct way to test whether our MK-801 or D-APV in vivo administration blocked or reduced Ca^2+^ influx of NMDAR, is doing calcium imaging or electrical recording on brain slides after acute and chronic MK-801 or D-APV administration. However, during the pandemic of COVID-19, it is difficult to extend our studies to perform this experiment currently, however, we believe the literature is fairly clear that our treatments are surely affecting NMDAR function profoundly. We do plan to pursue experiments like this in the future.

The drugs blocking NMDAR activity lack sufficient controls for the conclusions stated in the manuscript: Loss of the innervation pattern of innervation is variable and the S1/S2 column indicates a possible dependency on activity, leading to other possible interpretations. Changes in the overall cortical activity may affect the stabilization of callosal projections, and therefore acute pharmacological blocking of NMDAR may not mimic the KO or antibody treatments. Early and permanent loss of NMDAR activity can trigger changes in the intrinsic excitability of the neurons that could override the phenotypes of increased innervation. This could be discussed textually

We agree and appreciate the valuable advice. In the newly edited manuscript, we disused these possibilities in our Discussion “Previous studies have observed the role of NMDAR in axon elimination (Colonnese and Constantine-Paton, 2006; Henson et al., 2017; Personius et al., 2016; Rabacchi et al., 1992; Rajan et al., 1999; Zhang et al., 2013). in vivo time-lapse images of retinal axons from albino *Xenopus* tadpoles show that correlated neural activity influences axon branch dynamics of retinal axons in the optic tectum (Ruthazer et al., 2003). The axon branches whose activity is not correlated with that of their neighbors are selectively eliminated. However, the selective elimination can be abolished by blocking the Ca^2+^ influx of NMDAR by its antagonist MK-801. When we blocked the Ca^2+^ influx of NMDAR by MK-801 or D-APV, even though the overall callosal innervation pattern of somatosensory cortex was not affected, there were some subtle, local laminar-specific innervation differences at the S1/S2 border, suggesting that the refinement of axon arborization may rely on the Ca^2+^ influx-mediated activity changes through NMDAR” when mentioned about “The refinement of callosal innervation pattern”.

For the Eph/ephrin data:3) You argue EphB2 protein levels are reduced in the cortex of conditional NR1 floxed mice and disruption of ephrin-B/EphB signaling in this system leads to similar termination defects as deleting NR2B. This is an apt argument but the reviewers felt that this was only a first step in the required analyses.

We have toned down the linkage between NMDAR and EPHRINB/EPHB in the Abstract, the Results and the Discussion. We also removed the linkage between NMDAR and EPHRINB/EPHB in the title. We will examine how and whether NMDAR regulates EPHRIN-B/EPHB signaling directly and the detailed molecular mechanism underlying their collaboration to regulate callosal innervation patterns in the primary somatosensory cortex in a following project.

Reviewer #3:4) Figure 8. By Western blot analysis, the reduction seems 30% compared to controls. This is not dramatic and instead suggests other mechanisms at work, since most heterozygous mutants (with target protein reductions at 50% or more) do not show phenotypes.5) Partial reduction of EphB2 protein raises the possibility that further reductions of EphB2 signaling may enhance the phenotype. Instead, blocking NMDAR plus EphB2 functions (by injecting NR1Ab and ephrin-B1-Fc) did not enhance the phenotype of the single manipulations. One would expect additive effects.

Questions 4 and 5 are interlinked, so we are addressing them together. We understand the reviewer’s concern that a 30% reduction of EPHB2 in S1 of *Emx1^cre/+^*; *Grin1^fl/fl^* mice by western blot was not dramatic. However, implicit in our studies was our assumption that the loss of EPHB2 would be selective for neurons that lost NMDAR and that this was a subset of cortical neurons. Also, we assumed that EPHB2 expressed in other compartments than the PSD might not be decreased. Thus, we believe that a 30% decrease is actually quite dramatic if the loss is limited to the PSD. We appreciate the reviewer asking us to more directly address this! *Emx1^cre/+^*; *Grin1^fl/fl^* mice only knocked out GluN1 in *Emx1^cre/+^* positive cells which were only a subpopulation cells in the cortex (excitatory neurons with Cre penetrance). We found that there are still many cells expressiong GluN1 in *Emx1^cre/+^* unaffected cell populations, such as interneurons (Figure 1—figure supplement 1). Also, to more specifically examine the expression of EPHB2 on *Emx1^cre/+^*; *Grin1^fl/fl^* positive cells, we crossed *Emx1^cre/+^*; *Grin1^fl/fl^* mice with Cre reporter mice-*Rosa26<fs-tdTomato^fl/fl^* mice to generate *Emx1^cre/+^*; *Grin1^fl/fl^; Rosa26<fs-tdTomato^fl/fl^*mice. We found the expression of EPHB2 was completely gone from somatic post-synaptic densities on *Emx1^cre/+^*; *Grin1^fl/fl^; Rosa26<fs-tdTomato^fl/fl^*positive cells. Thus, we believe that a loss of 30% of total EPHB2 protein in the cortex is actually quite dramatic. Importantly, this can’t be compared to EPHB2+/- mice where presumably any lost protein would be distributed throughout all domains and in all cells where the protein is expressed.

We also add the following paragraph into the newly edited text “As discussed above, in *Emx1^cre/+^*; *Grin1^fl/fl^* mice, GluN1 was selectively deleted in excitatory neurons. Based on previous studies, the expression of GluN1 in interneurons (Korotkova et al., 2010), oligodendrocytes and oligodendrocyte precursor cells (Karadottir et al., 2005) in the cortex should not be affected in *Emx1^cre/+^*; *Grin1^fl/fl^*mice. To examine the expression of EPHB2 specifically in GluN1 KO cells, we crossed *Emx1^cre/+^*; *Grin1^fl/fl^*mice with Cre-reporter mice – *Rosa26<fs-tdTomato^fl/fl^*mice that exhibit tdTomato expression after Cre-mediated recombination. Supporting our hypothesis that loss of GluN1 leads to loss of EPHB2, the punctate staining of EPHB2 on cell membranes was completely lost in red (recombined) cells of *Emx1^cre/+^*; *Grin1^fl/fl^*; *Rosa26<fs-tdTomato^fl/fl^* brain sections but not red cells of *Emx1^cre/+^*; *Grin1^wt/wt^;Rosa26<fs-tdTomato^fl/fl^* brain sections (Figure 8—figure supplement 2). Taken together, we demonstrated that the loss of NMDAR caused the loss of EPHB2 selectively on cells that lack GluN1 after excision, thus explaining the 30% reduction in EPHB2 protein expression. Given the known physical association between NMDAR and EPHB2, these data suggest reciprocity in this stabilizing interaction and dendritic localization” when mentioned about “EPHB2 protein expression is decreased in GluN1 KO mice”.

6) Figure 8, only the timepoint P6 is shown. Rev 5 also points out that between P6 and P10 there is more variability in contralateral innervation correlating with the anterior-posterior axis. You should show ephrin B electroporations analyzed at P14-P30, or remove and/or tone down the text.

We have toned down the discussion of the interaction between NMDAR and EPHRINB/EPHB.

Reviewer #4:7) EphB-ephrin-B signaling and NMDAR function likely has to do with EphB-ephrin-B repulsive cues or that EphBs interact and modulate NMDAR function, and regulate excitatory synapse development in forebrain. It seems at least equally likely that in the latter case, EphB modulation of NMDAR may mediate the events described in this manuscript, on synapse development and maintenance, not necessarily on repulsion/guidance.

We agree and thank the reviewer for these suggestions. We have discussed these alternative possibilities in the newly edited manuscript. We have added the following paragraph into our Discussion “Regarding previous roles of EPHRIN-B/EPHB signaling in pruning, this pathway was previous shown to be involved in infrapyramidal bundle pruning in dentate gyrus (Xu and Henkemeyer, 2009) and it is know that activation of EPHB by EPHRIN-B potentiates Ca^2+^ influx of NMDA receptor (Takasu et al., 2002). These lines of evidence suggest that NMDAR and EPHRIN-B/EPHB signaling may cooperate in the refinement of the callosal innervation pattern and perhaps in other circuits. Further studies will examine how and whether NMDAR regulates EPHRIN-B/EPHB signaling directly and the detailed molecular mechanism underlying their collaboration to regulate callosal innervation patterns in the primary somatosensory cortex” when discussed “the refinement of callosal innervation pattern”.

8) EphB-NMDAR interaction results in phosphorylation of GluN2B, increased synaptic localization of the NMDAR, and increased NMDAR dependent Ca influx. Loss of EphB-ephrin-B signaling might cause a reduction in NMDAR Ca influx, but not a complete block of ionotropic signaling. To demonstrate that the findings in Figures 7 and 8 are linked, the authors should inject NMDAR blockers (after conducting the control experiments in point 2), in the ephrin-B1 knockout mice. The anti-NMDAR antibodies simply remove the receptor from the synapse. This does not address whether the effects are independent of the ionotropic modulation of the channel and should be discussed.

We agree and think this is a reasonable point. Our toned down discussion of the interaction hopefully will address this comment. We will perform this very sensible experiment when addressing the detailed molecular mechanism underlying the collaboration of NMDAR and EPHRINB/EPHB signaling in regulating callosal innervation patterns in the following project.

Revisions expected in follow-up work:1) In implicating repellent activity that would restrict commissural axonal projections to the S1 cortex, the evidence for NMDAR via Eph B2 function as repellents for elongating callosal axons is lacking. in vitro experiments demonstrating repellent activity of EphB downstream of NMDAR would be necessary to make this claim. Given the time required to develop and execute such an assay, and the pandemic delays, you could take as long as needed to do this experiment, if you choose to do so.

We will address it in our follow-up project.

2) Reviewers 4 and 5 point out that these pathways could be involved in refinement and pruning of axons: Reviewer 5 raises the possibility that in the mutants, axons are stabilized, to result in extra numbers of callosal neurons. NMDAR could thus be responsible for repulsion during refinement. Reviewer 4's comments on EphBs interacting with excitatory synapse development could play into this explanation. More high-resolution analysis of the impact of your manipulations on axonal branching would address the other reviewers' questions regarding whether the changes see are due defects in branching and/or refinement. This aspect as in the in vitro experiments require additional experiments which could understandably not be possible at this time.

Thank you for your kindly considerations. We appreciate it. We have discussed all the possibilities raised by reviewers 4 and 5 in our last part of Discussion -“the refinement of callosal innervation pattern”. As for the high-resolution of individual axon branch raised by reviewer 3, we do appreciate the advice and have applied sparse labelling of axon branch in another project when addressing the effect of patient-derived anti-NMDAR autoantibodies on callosal innervation pattern in S1. We are also planning to sparsely label neurons in vivo when addressing the detailed molecular mechanism underlying the collaboration of NMDAR and EPHRINB/EPHB signaling in regulating callosal innervation patterns in the follow-up project. We do hope we can contribute more to the field in these follow up studies.

[Editors' note: further revisions were suggested prior to acceptance, as described below.]

Revisions:The rebuttal letter was difficult to follow. It was not clear where in the manuscript some of the changes were made. Figures were not labelled and main figures were mixed with supplementary figures. However, the following points appear not to have been addressed adequately:1) In response to the criticism of overstating the 30% reduction in EphB2 protein expression in the Grin1 CKOs, the authors have provided images showing EphB2 immunofluorescence in Emx1Cre;Grin1fl/fl;tdTomato mice (Figure 8—figure supplement 2). While the pictures in this new figure suggest loss of EphB2 IF in Tom+;Grin1 cKO cells, but not in Tom+;control cells, the results should be quantified. Presenting images without quantification is not convincing. Please quantify and add the results to the paper.

We have quantified the fluorescence density of EPHB2 immunostaining for each cell. See the Figure 8—figure supplement 2.

2) Please address the following points that were raised in the previous critique and appear not to have been answered:a) show images of all controls and experimental groups of Figure 3 in Supplementary data.

The images of the rest of controls and experimental groups for double in utero electroporation have been added it into manuscript as Figure 3—figure supplement 1.

b) Provide better evidence for NR1 protein downregulation in the conditional NR1 knockouts (Figure 1—figure supplement 1).

In addition to the immunostaining data, we performed western blot of S1 at P8. Compared with control, the protein levels of GluN1 were greatly reduced in *Emx1^cre/+^*; *Grin1^fl/fl^* mice, see Figure 1—figure supplement 1.